# Self-Play $Q$-Learners Can Provably Collude in the Iterated Prisoner's Dilemma

**Quentin Bertrand** [1]   **Juan Agustin Duque** [2]   **Emilio Calvano** [3]   **Gauthier Gidel** [2 4]

## Abstract

A growing body of computational studies shows that simple machine learning agents converge to cooperative behaviors in social dilemmas, such as collusive price-setting in oligopoly markets, raising questions about what drives this outcome. In this work, we provide theoretical foundations for this phenomenon in the context of self-play multi-agent Q-learners in the iterated prisoner's dilemma. We characterize broad conditions under which such agents provably learn the cooperative Pavlov (win-stay, lose-shift) policy rather than the Pareto-dominated "always defect" policy. We validate our theoretical results through additional experiments, demonstrating their robustness across a broader class of deep learning algorithms.

## 1. Introduction

In recent years, algorithmic pricing has increasingly supplanted manual pricing, both online—by 2015, one-third of Amazon's largest third-party sellers were already using such tools (Chen et al., 2016)—and in physical markets, such as gas stations (Schechner, 2017). The growing adoption of machine learning–based pricing systems has raised significant concerns in both academic (Ezrachi & Stucke, 2015; Mehra, 2015) and institutional (OECD, 2017; Bureau, 2018) circles regarding the potential for tacit collusion. Indeed, if multiple firms competing within the same local market deploy learning algorithms to set prices, a natural and pressing question arises:

*Could pricing algorithms autonomously learn to suppress competition, thereby leading to higher prices?*

Prior works—based on simulated environments and empirical analyses of real-world pricing—have shown that algorithmic collusion is a tangible risk, not merely a theoretical concern. For instance, Deneckere & Davidson (1985); Waltman & Kaymak (2008); Hansen et al. (2021); Asker et al. (2022) document that even simple algorithms can lead to higher prices in standard models of price competition. Moreover, Calvano et al. (2019; 2020b); Klein (2021) demonstrate that standard machine learning algorithms, such as $Q$-learning (Watkins & Dayan, 1992), can learn to sustain high prices through reward–punishment mechanisms—patterns commonly characterized as 'collusive' in the antitrust literature; see Calvano et al. 2020a for an in-depth discussion of related policy issues. Assad et al. (2024) go one step further by providing the first empirical evidence of how algorithmic pricing affects competition: in Germany's retail gasoline market, where machine learning pricing tools spread after 2017, adoption increased profit margins only when multiple competitors used them—consistent with concerns over algorithm-driven collusion. More broadly, cooperation among machine learning algorithms has also been observed in general iterated social dilemmas, owing to the sequential nature of the problems (Lanctot et al., 2017; Leibo et al., 2017).

Theoretically, the emergence of tacit collusion has been studied in highly simplified settings. Even in minimalist cooperative/competitive games such as the prisoner's dilemma (Flood, 1958; Kendall et al., 2007), the dynamics induced by standard machine learning algorithms in multi-agent contexts are often so complex that researchers typically resort to oversimplified versions of these algorithms. In particular, expected-value formulations of $\epsilon$-greedy $Q$-learning (Seijen et al., 2009; Sutton & Barto, 2018, Sec. 6.10), *without memory*—that is, without access to previous states—are frequently analyzed (Banchio & Skrzypacz, 2022; Banchio & Mantegazza, 2022). In these prior theoretical results, cooperation does not arise from agents learning equilibrium strategies in the classical game-theoretic sense. Their limited memory prevents them from conditioning on the past actions of others, and thus from learning to implement retaliatory pricing strategies (Calvano et al., 2023). By contrast, learning to sustain high prices through retaliation consistent with a subgame-perfect equilibrium represents a stronger and more strategic form of cooperation: it captures

---
[*]Equal contribution  [1]Université Jean Monnet Saint-Etienne, CNRS, Institut d'Optique Graduate School, Inria, Laboratoire Hubert Curien UMR 5516, F-42023, Saint-Étienne, France [2]Mila, Université de Montréal  [3]Università LUISS (Rome), Toulouse School of Economics, EIEF and CEPR  [4]Canada AI CIFAR Chair.  Correspondence to: Quentin Bertrand <quentin.bertrand@inria.fr>.

*Proceedings of the 42$^{nd}$ International Conference on Machine Learning*, Vancouver, Canada. PMLR 267, 2025. Copyright 2025 by the author(s).

equilibrium behavior that is robust to unilateral deviations —unlike the limited-memory scenarios, where cooperation can emerge without credible deterrence mechanisms.

In this paper, we aim to fill this gap by studying a richer setting that, in principle, allows for collusion in the game-theoretic sense. More precisely, the proposed project seeks to shed light on the *learning dynamics of algorithms* as they coordinate to cooperate—*in contrast to the learning outcomes*, which have been the primary focus of prior work. By analyzing the processes that lead to collusion in a minimalist *self-play* setting—*i.e.,* where an agent interacts with a copy of itself—we aim to better inform the development of rules and regulations that ensure fair competition and protect consumer interests.

**Contributions.** In this work, we study the *dynamics* of two agents playing the iterated prisoner's dilemma and choosing their actions according to *self-play* $\epsilon$-greedy $Q$-learning policies. Importantly, as opposed to previous work (Banchio & Skrzypacz, 2022; Banchio & Mantegazza, 2022), we theoretically study the standard *stochastic* (*i.e.,* not averaged) version of *self-play* $\epsilon$-greedy $Q$-learning (Algorithm 1), *with a one-step memory*. More precisely,

- First, we show that without exploration, *i.e.,* self-play $\epsilon$-greedy $Q$-learning with $\epsilon = 0$, if the initialization of the agent is optimistic enough (Even-Dar & Mansour, 2001), then agents can learn to move from an `always defect` policy, to a cooperative policy (Theorem 3.2 in Section 3.1), referred to as `win-stay, lose-shift` policy (Nowak & Sigmund, 1993).
- Then, we extend the convergence toward a cooperative policy to self-play $\epsilon$-greedy $Q$-learning with $\epsilon > 0$ (Theorem 3.3 in Section 3.2). This is the main technical difficulty, as one needs to prove the convergence of a stochastic process toward a specific equilibrium.
- Finally, we empirically show that the collusion proved for standard $Q$-learning algorithms is also observed for deep $Q$-learning algorithms (Section 5).

The manuscript is organized as follows: Section 2 provides recalls on the prisoner's dilemma, multi-agent $Q$-learning, and fixed points of the self-play multi-agent Bellman equation. Section 3 contains our main results: the convergence of $Q$-learning toward the collusive `Pavlov` strategy (Theorems 3.2 and 3.3). Previous related works on the dynamics of multi-agent $Q$-learning are detailed in Section 4. Similar collusive behaviors are also observed for deep $Q$-learning algorithms in Section 5.

## 2. Setting and Background

First, Section 2.1 provides recalls on the iterated prisoner's dilemma. Then, Section 2.2 provides recalls on reinforcement learning, and more specifically, multi-agent Bellman

equation and $\epsilon$-greedy $Q$-learning. In Section 2.3, we recall the fixed-point policies of the multi-agent Bellman equation: `always defect`, `Lose-shift`, `Pavlov` (summarized in Table 2). Interestingly, in the iterated prisoner's dilemma, these policies are also *subgame perfect equilibrium*, and might be different from `always defect`.

### 2.1. Iterated Prisoner's Dilemma

In this work, we consider a two-player iterated prisoner's dilemma whose players have a *one-step* memory. At each time step, players choose between cooperate (C) or defect (D), *i.e.,* each player $i$ choose an action $a^i \in \mathcal{A} \triangleq \{C, D\}$, which yields a reward $r^i_{a^1, a^2}$ for player $i$, $i \in \{1, 2\}$. Table 1 describes the rewards $r^1_{a^1, a^2}$ and $r^2_{a^2, a^1}$ respectively obtained by each player depending on their respective action $a^1$ and $a^2$. Note that such a game is *symmetric*: $r^1_{a^1, a^2} = r^2_{a^2, a^1} := r_{a^1, a^2}$. For simplicity, in the experiments (Section 5), we consider simplified rewards, which are parameterized by a single scalar $g$, $1 < g < 2$ (see Table 3 in Appendix E, as in Banchio & Mantegazza 2022).

When the prisoner's dilemma is not repeated, joint defection is the only Nash equilibrium: for a fixed decision of the other prisoner, defecting always reaches better rewards than cooperating. This yields the celebrated paradox: even though the rewards obtained in the "defect defect" state are Pareto dominated by the ones obtained with the "cooperate cooperate" state, the Pareto-suboptimal mutual defection remains the only Nash equilibrium.

When the prisoner's dilemma is infinitely repeated, new equilibria can emerge, and `always defect` is no longer the dominant strategy (Osborne, 2004). We will refer to "collusive" or "colluding" strategies, any strategy that differs from the `always defect` strategy. Note that infinitely many time steps are essential for such equilibria to exist[1].

*Table 1.* Prisoner's Dilemma Rewards/Payoff Matrix. Typical iterated prisoner's dilemma rewards satisfy $r_{\text{DC}} > r_{\text{CC}} > r_{\text{DD}} > r_{\text{CD}}$ and $2r_{\text{CC}} > r_{\text{CD}} + r_{\text{DC}}$.

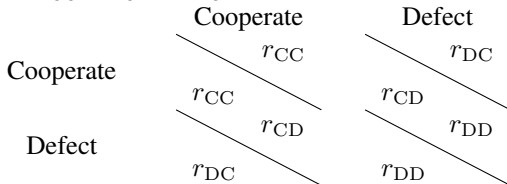

---

[1]Equivalently, one can consider that the game is finitely repeated with a probability of ending equal to the discount factor at each step (Littman, 1994; Osborne, 2004, Chap. 15.3).

**Algorithm 1** Multi-agent Self-Play $Q$-learning

**init**  : $s_0, Q$
**param** : $\alpha, \gamma, \epsilon$
**for** $t$ *in* $1, \ldots, n_{\text{iter}}$ **do**
   // Compute $a_t^1$ and $a_t^2$ via Algorithm 2
   $a_t^1 = $ Algorithm $2(Q, (a_{t-1}^1, a_{t-1}^2), \epsilon)$
   $a_t^2 = $ Algorithm $2(Q, (a_{t-1}^2, a_{t-1}^1), \epsilon)$
   $s_t = (a_{t-1}^1, a_{t-1}^2)$
   // Update the $Q$-value entry in $s_t, a_t^1$
   $Q_{s_t, a_t^1} \mathrel{+}= \alpha \left( r_{a_t^1, a_t^2} + \gamma \max_a Q_{(a_t^1, a_t^2), a} - Q_{s_t, a_t^1} \right)$
**return** $Q$

## 2.2. $Q$-Learning for Multi-Agent Reinforcement Learning and Self-Play

**Reinforcement learning**. At a given step, the choice of each player to cooperate or defect is conditioned by the actions $\mathcal{S} \triangleq \{\text{CC}, \text{DD}, \text{CD}, \text{DC}\}$ played at the previous time step, where the first action stands for the action picked by the first player at the previous time-step. $\Delta_{\mathcal{A}}^{\mathcal{S}}$ denotes the space of policies $\pi : \mathcal{S} \times \mathcal{A} \to [0, 1]$, such that for all $s \in \mathcal{S}$, $\sum_{a \in \mathcal{A}} \pi(a|s) = 1$. Given two policies (*i.e.,* mixed strategies) $\pi_1, \pi_2 \in \Delta_{\mathcal{A}}^{\mathcal{S}}$, a discount factor $\gamma \in (0, 1)$, and an initial distribution $\rho$ over the state space $\mathcal{S}$, the cumulated reward observed by the agent $i \in \{1, 2\}$ is given by

$$J_i(\pi_1, \pi_2) \triangleq \mathbb{E}_{s_0 \sim \rho, a_t^i \sim \pi_i(\cdot|s_t)} \left[ \sum_{t=0}^{\infty} \gamma^t r_{a_t^1, a_t^2} \right] \ . \quad (1)$$

$Q$**-learning**. A popular approach to maximize the cumulative reward function Equation (1) is to find an *action-value function* or *Q-function*, that is a fixed point of the Bellman operator (Sutton & Barto, 2018, Eq. 3.17). Since the reward of the second player is stochastic in the multi-agent case, the fixed-point equation writes, for all $(s_t, a_t^1) \in \mathcal{S} \times \mathcal{A}$:

$$Q_{s_t, a_t^1}^\star = \mathbb{E}_{a_t^2 \sim \pi_2(\cdot|s_t)} \left( r_{a_t^1, a_t^2}^1 + \gamma \max_a Q_{(a_t^1, a_t^2), a}^\star \right) \ . \quad (2)$$

$Q$-learning (Algorithm 1) consists of stochastic fixed-point iterations on the Bellman Equation (4): with a step-size $\alpha > 0$

$$\begin{aligned} Q_{s_t, a_t^1}^{t+1} = &\; Q_{s_t, a_t^1}^t \\ &+ \alpha \left( r_{a_t^1, a_t^2}^1 + \gamma \max_{a'} Q_{(a_t^1, a_t^2), a'}^t - Q_{s_t, a_t^1}^t \right) \ . \end{aligned} \quad (3)$$

**Self-play**. Since the game is symmetric, we study the dynamic of Equation (3), where the second agent is a copy of the first agent, *i.e., the actions* $a_t^1$ *and* $a_t^2$ *are sampled according to the same policy* $\pi$. Such a way to model the opponent is referred to as *self-play* and has been successfully used to train agents in deep reinforcement

learning applications (Lowe et al., 2017; Silver et al., 2018; Baker et al., 2019; Tang, 2019):

$$a_t^1, a_t^2 \sim \pi(\cdot|s_t) \quad \text{// same } Q\text{-table for } a_t^1 \text{ and } a_t^2.$$

In the self-play setting, the Bellman Equation (2) writes

$$Q_{s_t, a_t^1}^\star = \mathbb{E}_{a_t^2 \sim \pi(\cdot|s_t)} \left( r_{a_t^1, a_t^2} + \gamma \max_a Q_{(a_t^1, a_t^2), a}^\star \right) . \quad (4)$$

In addition, we study $\epsilon$-greedy $Q$-learning policies (Algorithm 2, $0 < \epsilon < 1/2$), *i.e.,*

$$\pi(a|s) = \begin{cases} 1 - \epsilon & \text{if } a = \arg\max_a Q_{s,a} \\ \epsilon & \text{else} \end{cases}, i \in \{1, 2\},$$

We say that an $\epsilon$-greedy policy is a fixed point of the self-play multi-agent Bellman Equation (4) if the policy is greedy with respect to its own $Q$-values. In other words, a fixed point $Q^\star$ of the self-play multi-agent Bellman Equation (4) must satisfy Equation (4), where the policy $\pi$ is $\epsilon$-greedy with respect to the $Q$-table $Q^\star$. Interestingly, in this setting, there exist multiple fixed-point policies $Q^\star$ of the self-play multi-agent Bellman Equation (4).

In this paper, we study the dynamics of the multi-agent self-play $Q$-learning (Equation (3)) in the *one-step memory* case, *i.e.,* $\mathcal{A} = \{\text{C}, \text{D}\}$ and $\mathcal{S} = \{\text{C}, \text{D}\}^2$. In particular, in Section 3 we show convergence towards a specific cooperative fixed point policy $Q^\star$ of the multi-agent Bellman Equation (4). In the following Section 2.3, we provide recalls on the fixed-point policies for the multi-agent Bellman equation Equation (4).

### 2.3. Fixed Points of the Multi-Agent Bellman Equation

In the iterated prisoner's dilemma with one-step memory, multiple symmetric strategy profiles exist. However, only three of them are fixed points of the self-play multi-agent Bellman operator in Equation (4): `always defect`, `Pavlov`, and `grim trigger` (Usui & Ueda, 2021). A summary of these policies is provided in Table 2.

In Section 3, we show that one can transition from `always defect` to `Pavlov`, passing through the `lose-shift` policy (defined in Table 2). First, we recall below that fixed-point policies of the multi-agent Bellman Equation (4) can define `always defect` and `Pavlov`.

**Proposition 2.1** (`Always Defect`). *The following $Q$-table $Q^{\star, \text{Defect}}$ is a fixed point of the Bellman Equation (4) and yields an* `always defect` *policy,* $\forall s \in \{\text{C}, \text{D}\}^2$,

$$Q_{s, \text{D}}^{\star, \text{Defect}} = \mathbb{E}_{a^2 \sim \pi(\cdot|s)} r_{\text{D}, a^2} / (1 - \gamma) \ ,$$
$$Q_{s, \text{C}}^{\star, \text{Defect}} = Q_{s, \text{D}}^{\star, \text{Defect}} - \mathbb{E}_{a^2 \sim \pi(\cdot|s)} \left( r_{\text{D}, a^2} - r_{\text{C}, a^2} \right) \ .$$

**Algorithm 2** $\epsilon$-greedy

---

**input** : $Q, s, 0 \leq \epsilon \leq 1/2$
**return** $\arg\max_{a'} Q_{s,a'}$ with probability $1 - \epsilon$ and $\arg\min_{a'} Q_{s,a'}$ with probability $\epsilon$

---

More interestingly, the cooperative `Pavlov` policy (also referred to as `win-stay, lose-shift`) can also be a fixed point of the multi-agent Bellman Equation (4). `Pavlov` can be summarized as *cooperate as long as the players are synchronized by playing the same action.*

**Proposition 2.2** (`Pavlov`). *If $\gamma > \frac{r_{DC} - r_{CC}}{r_{CC} - r_{DD}}$ and $\epsilon$ is small enough, then there exists a Q-function, $Q^{\star,\mathrm{Pavlov}}$, which is a fixed point of the self-play multi-agent Bellman Equation* (4) *and yields the Pavlov policy, i.e,*

$$\forall s \in \{CC, DD\} \quad Q_{s,C}^{\star,\mathrm{Pavlov}} > Q_{s,D}^{\star,\mathrm{Pavlov}} \quad and$$
$$\forall s \in \{CD, DC\} \quad Q_{s,C}^{\star,\mathrm{Pavlov}} < Q_{s,D}^{\star,\mathrm{Pavlov}} \quad.$$

Proofs of Propositions 2.1 and 2.2, the exact $Q$-values and condition on $\epsilon$ are recalled in Appendices A.1 and A.2 for completeness. In the rest of the manuscript we assume that $\gamma > \frac{r_{DC} - r_{CC}}{r_{CC} - r_{DD}}$, and the exploration $\epsilon$ is small enough such that `Pavlov` policy exists.

The main takeaway from Proposition 2.2 is that there exists a fixed point of the self-play multi-agent Bellman Equation (4) whose associated strategy is cooperative. Interestingly, the only other fixed-point strategy is the (cooperative) `grim trigger` policy (Usui & Ueda 2021, Table 1; Meylahn & Janssen 2022). On the opposite, *tit-for-tat* is not a fixed-point policy of the Bellman Equation (4). Table 2 summarizes the greedy action of each fixed-point policy. Interestingly, `always defect`, `Pavlov`, and `grim trigger` policies also are *subgame perfect equilibrium*, which is a stronger notion of equilibrium than Nash equilibrium for iterated games (the subgame perfect definition can be found in Sec. 5.5 of Osborne 2004). For completeness, the proofs of the latter statement are recalled in Appendix B.

Propositions 2.1 and 2.2 illustrate the challenges of multi-agent reinforcement learning for the prisoner's dilemma: multiple equilibria exist that yield outcomes with markedly different levels of cooperation. Thus, one should not only care about convergence toward equilibrium, but one should also care about *which type* of equilibrium the agents converge to. The reached equilibrium will depend on the *dynamics* of the algorithm used to estimate the policy $\pi^\star$: this dynamics is studied in Section 3 for $Q$-learning.

*Table 2.* Summary of the symmetric fixed-point policies of the self-play multi-agent Bellman Equation (4). The table displays the greedy action $a_t$ of each policy, given the state $s_t = (a_{t-1}^1, a_{t-1}^2)$.

| State $s_t$ / Policy | (D, D) | (C, C) | (C, D) | (D, C) | Fixed Point |
|---|---|---|---|---|---|
| Always defect | D | D | D | D | ✓ |
| Lose-shift | C | D | D | D | ✗ |
| Grim trigger | D | C | D | D | ✓ |
| Pavlov | C | C | D | D | ✓ |

## 3. $Q$-Learning Dynamic in the Iterated Prisoner's Dilemma

The complex structure of the multi-agent $Q$-learning with memory yields multiple fixed-point policies for the Bellman Equation (4). In this section, for a specific set of $Q$-value initializations $Q^0$, we show convergence of the dynamics resulting from $\epsilon$-greedy $Q$-learning with memory updates (*i.e.,* Equation (3)) toward the cooperative fixed point `Pavlov` policy.

More precisely, we show that, with an "optimistic enough" initialization, the $\epsilon$-greedy policy with a small enough learning rate $\alpha$ starts from the `always defect` policy, then moves to the `lose-shift` policy, and finally converges towards the `Pavlov` policy. The $Q$-values at initialization are required to be "optimistic enough" in the sense that they are "set to large values, larger than their optimal values" (Even-Dar & Mansour, 2001).

For exposition purposes, the greedy case ($\epsilon = 0$ in Algorithm 2) is presented in Section 3.1, and the general case ($0 < \epsilon < 1/2$) in Section 3.2.

### 3.1. Fully Greedy Policy with No Exploration (Algorithm 2 with $\epsilon = 0$)

We first show the case $\epsilon = 0$, *i.e.,* with no exploration. We assume that the initial policy is `always defect`, and we require "optimistic enough" initial $Q$-values.

**Assumption 3.1** ($Q$-values Initialization).

*i)* $\frac{r_{DD}}{1-\gamma} < Q_{(D,D),C}^{t_0}$ ,

*ii)* $Q_{(D,D),C}^{t_0} < \frac{r_{CC}}{1-\gamma} - \frac{r_{CC} - r_{DD}}{1-\gamma^2} < Q_{(C,C),C}^{t_0}$ and $Q_{(D,D),D}^{t_0} < Q_{(C,C),C}^{t_0}$ .

*iii)* $Q_{(C,C),C}^{t_0} < \frac{r_{CC}}{1-\gamma}$ .

Assumption 3.1 *i)* ensures that the algorithm moves from `always defect` to `lose-shift` policy. Assumption 3.1 *ii)* ensure that the algorithm shifts from `lose-shift` to `Pavlov` policy, and Assumption 3.1 *iii)* ensures that the policy sticks to `Pavlov`.

*$Q$-values Initialization in Practice*. How can we ensure

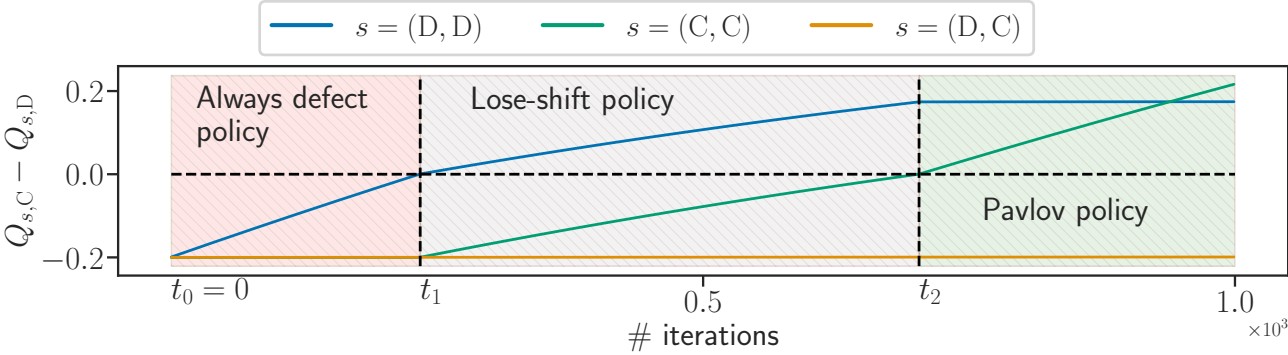

*Figure 1.* **From `always defect` to `Pavlov` policy, Algorithm 1 with no exploration** (*i.e.,* $\epsilon = 0$). Evolution of the $Q$-learning policy as a function of the number of iterations in the iterated prisoner's dilemma. With a correct (optimistic) initialization, players go from an `always defect` policy to the `lose-shift` policy (at time $t_1$), and then go to the cooperative `Pavlov` policy (at time $t_2$). The incentive to cooperate and the discount factor are set to $g = 1.8$ (see Table 3) and $\gamma = 0.6$.

that Assumptions 3.1 *i)* to 3.1 *iii)* are actually satisfied in practice? For standard $Q$-learning, the initial $Q$-values are user-defined parameters of the algorithm (see Algorithm 1). Figure 2 illustrates how the initial $Q$-values—specifically Assumptions 3.1 *i)* and 3.1 *ii)*—influence the resulting policy. However, in more complex settings such as deep $Q$-learning, the initial $Q$-values are not explicitly controlled, since they result from randomly initialized neural network weights. In such cases, initializing the weights to directly satisfy Assumption 3.1 is non-trivial.

To address this, we propose a practical approach to approximate initialization toward an `always defect` policy, which can subsequently shift toward cooperation. Specifically, we suggest initializing the $Q$-values using the output of Algorithm 1 executed with an exploration parameter $\epsilon = 1/2$. This corresponds to a uniformly random policy: $p(C \mid s) = p(D \mid s) = \frac{1}{2}, \quad \forall s \in \{CC, DD, CD, DC\}$. Under this initialization, the self-play multi-agent Bellman Equation (4) admits a unique fixed-point policy, which corresponds to the `always defect` strategy.

**Theorem 3.2.** *Suppose the initial policy is* `always defect` *and the initial state* $s_0$ *is* defect defect: $s_0 = \mathrm{DD}$. *Then for all* $Q$-values *initializations* $Q^{t_0}$ *that satisfy Assumption 3.1, Algorithm 1 with no exploration* ($\epsilon = 0$) *moves away from the* `always defect` *policy, and learn the cooperative* `Pavlov` *policy.*

Figure 1 illustrates Theorem 3.2 and shows the evolution of Equation (4) as a function of the number of iterations. At $t = t_0$, $Q_{s,\mathrm{D}} > Q_{s,\mathrm{C}}$ for all states $s \in \{C, D\}^2$. The greedy action is playing defect, $Q_{(\mathrm{D},\mathrm{D}),\mathrm{D}} - Q_{(\mathrm{D},\mathrm{D}),\mathrm{C}}$ increases and agents progressively learn the `lose-shift` policy. Once the `lose-shift` policy is learned, the greedy actions are "cooperate" when the state is DD and "defect" when the state is CC. Hence, in this phase, Algorithm 1 successively

updates $Q_{(\mathrm{C},\mathrm{C}),\mathrm{D}}$ and $Q_{(\mathrm{D},\mathrm{D}),\mathrm{C}}$ entries, until $Q_{(\mathrm{C},\mathrm{C}),\mathrm{D}}$ goes below $Q_{(\mathrm{C},\mathrm{C}),\mathrm{C}}$. From this moment, the dynamic changes, and agents start to play the `Pavlov` policy and do not change.

*Proof.* (Theorem 3.2) As illustrated in Figure 1, the trajectory can be decomposed in 3 phases: first, the policy goes from `always defect` to `lose-shift` policy (Phase 1). Then, the policy goes from `lose-shift` to `Pavlov` (Phase 2). Finally, the policy stays in the `Pavlov` policy (Phase 3).

● **Phase 1 – From `always defect` to `lose-shift`** ($0 \leq t \leq t_1$). At $t_0$ we have, $s_0 = \mathrm{DD}$, and for all $s \in \{\mathrm{C}, \mathrm{D}\}^2$, $Q_{s,\mathrm{D}} > Q_{s,\mathrm{C}}$, hence the greedy action is defect, and is always chosen since $\epsilon = 0$. Then, the $Q$-learning update Equation (3) writes

$$Q_{(\mathrm{D},\mathrm{D}),\mathrm{D}}^{t+1} = Q_{(\mathrm{D},\mathrm{D}),\mathrm{D}}^{t} + \alpha \left( r_{\mathrm{DD}} + \gamma Q_{(\mathrm{D},\mathrm{D}),\mathrm{D}}^{t} - Q_{(\mathrm{D},\mathrm{D}),\mathrm{D}}^{t} \right). \quad (5)$$

Thus, while $Q_{(\mathrm{D},\mathrm{D}),\mathrm{D}} > Q_{(\mathrm{D},\mathrm{D}),\mathrm{C}}$, $Q_{(\mathrm{D},\mathrm{D}),\mathrm{D}}$ is the only updated entry and converges linearly towards $Q_{(\mathrm{D},\mathrm{D}),\mathrm{D}}^{\star,\mathrm{Defect}}$:

$$Q_{(\mathrm{D},\mathrm{D}),\mathrm{D}}^{t+1} - Q_{(\mathrm{D},\mathrm{D}),\mathrm{D}}^{\star,\mathrm{Defect}}$$
$$= (1 - \alpha(1 - \gamma))^t \left( Q_{(\mathrm{D},\mathrm{D}),\mathrm{D}}^{t} - Q_{(\mathrm{D},\mathrm{D}),\mathrm{D}}^{\star,\mathrm{Defect}} \right),$$

where $Q_{(\mathrm{D},\mathrm{D}),\mathrm{D}}^{\star,\mathrm{Defect}} \triangleq r_{\mathrm{D},\mathrm{D}}/(1 - \gamma)$. Thus, since $Q_{(\mathrm{D},\mathrm{D}),\mathrm{C}}^{t_0} > Q_{(\mathrm{D},\mathrm{D}),\mathrm{D}}^{\star,\mathrm{Defect}} \triangleq r_{\mathrm{D},\mathrm{D}}/(1 - \gamma)$ (Assumption 3.1 *i)*), $Q_{(\mathrm{D},\mathrm{D}),\mathrm{D}}$ converges linearly towards $Q_{(\mathrm{D},\mathrm{D}),\mathrm{D}}^{\star,\mathrm{Defect}} < Q_{(\mathrm{D},\mathrm{D}),\mathrm{C}}^{t_0}$. Hence there exists $t_1$ such that $Q_{(\mathrm{D},\mathrm{D}),\mathrm{C}}^{t_1} = Q_{(\mathrm{D},\mathrm{D}),\mathrm{C}}^{t_0} > Q_{(\mathrm{D},\mathrm{D}),\mathrm{D}}^{t_1}$. Once this time $t_1$ is reached, the policy switches from `always defect` to `lose-shift`, and the update in Equation (5) no longer guides the dynamics.

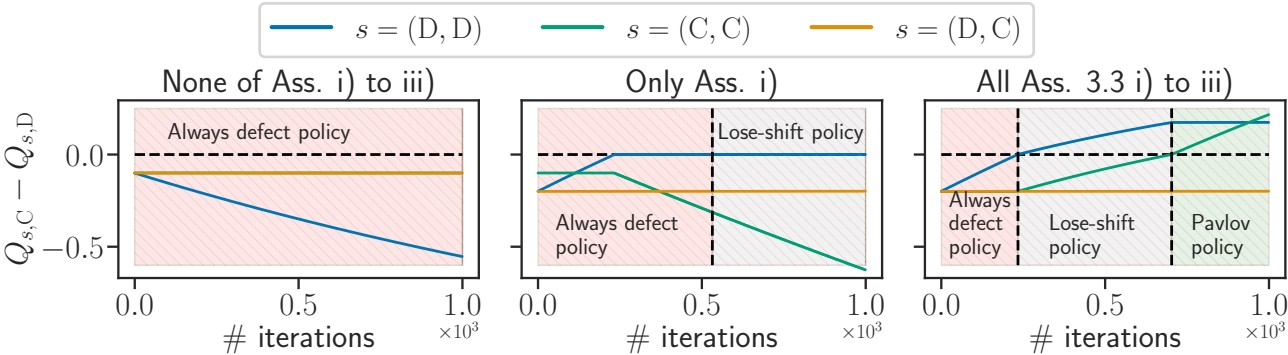

Figure 2. **Influence of Assumptions 3.1 i) to 3.1 iii)** ($\gamma = 0.6$ and $g = 1.8$ using the parameterization of Table 3). Evolution of the $Q$-learning policy as a function of the number of iterations in the iterated prisoner's dilemma, for multiple initializations. If one assumption from Assumptions 3.1 i) to 3.1 iii) is not satisfied, then the Pavlov policy is not achieved. Except for the initialization, the experimental setting is the same as for Figure 1.

• **Phase 2 – From lose-shift to Pavlov** ($t_1 \leq t \leq t_2$). In this phase, players alternate to defect in the CC state and cooperate in the DD state. Hence the only entries successively updated are $Q_{(C,C),D}$ and $Q_{(D,D),C}$. For all $t \geq t_1$, Equation (3) becomes

$$Q_{(C,C),D}^{2t+1} = (1-\alpha)Q_{(C,C),D}^{2t} + \alpha \left( r_{DD} + \gamma Q_{(D,D),C}^{2t} \right) ,$$
(6)

$$Q_{(D,D),C}^{2t+2} = (1-\alpha)Q_{(D,D),C}^{2t+1} + \alpha \left( r_{CC} + \gamma Q_{(C,C),D}^{2t+1} \right) .$$
(7)

Similarly to Phase 1, one can show that $Q_{(C,C),D}$ converges linearly towards $Q_{(C,C),D}^{\star,\text{lose-shift}}$. Additionally, one can show that, while the dynamic follows Equations (6) and (7), and $Q_{(C,C),D}^t > Q_{(C,C),C}^t$, then $Q_{(D,D),C}^t > Q_{(D,D),D}^t$ (see Lemma C.1 in Appendix C.1). Hence, there exists $t_2$ such that $Q_{(C,C),D}^{t_2} < Q_{(C,C),C}^{t_2}$ and $Q_{(D,D),C}^{t_2} > Q_{(D,D),D}^{t_2}$: the Pavlov policy is reached.

• **Phase 3 – Staying in Pavlov** ($t \geq t_2$).
In this part of the trajectory, both players cooperate in the state CC, and Equation (3) writes

$$Q_{(C,C),C}^{t+1} = Q_{(C,C),C}^t + \alpha \left( r_{CC} + \gamma Q_{(C,C),C}^t - Q_{(C,C),C}^t \right) .$$

The only updated $Q$-entry is $Q_{(C,C),C}$, and $Q_{(C,C),C}$ converges linearly toward $Q_{(C,C),C}^{\star,\text{Pavlov}}$

$$Q_{(C,C),C}^{t+1} - Q_{(C,C),C}^{\star,\text{Pavlov}}$$
$$= (1-\alpha(1-\gamma)) \left( Q_{(C,C),C}^t - Q_{(C,C),C}^{\star,\text{Pavlov}} \right) .$$

Since $Q_{(C,C),C}^{\star,\text{Pavlov}} \triangleq r_{CC}/(1-\gamma) > Q_{(C,C),D}^{t_2} = Q_{(C,C),D}^{t_0}$ (Assumption 3.1 ii), right), there is no other change of policy.

We have shown that the convergence is linear in each phase (from always defect to lose-shift, from lose-shift to Pavlov, staying in Pavlov). In addition, one can show that the time to go from one policy to another varies as $\mathcal{O}(1/\alpha)$ in each phase (see Appendix C.2 for the proof of this result). □

In Section 3.1, we showed that agents learned to cooperate with no exploration. In Section 3.2, we investigate cooperation in the general case, with an exploration parameter $\epsilon > 0$.

### 3.2. $\epsilon$-Greedy $Q$-Learning with Exploration (Algorithm 2 with $\epsilon > 0$)

In this section, we show that Algorithm 1 with sufficiently small step-size $\alpha$ and exploration parameter $\epsilon$ yields a cooperative policy with high probability.

**Theorem 3.3.** *Let $\delta > 0$ and $1/2 > \epsilon > 0$, $\epsilon$ sufficiently small such that Pavlov policy is a fixed-point of Equation (4) (as defined in Proposition 2.2). Suppose that Assumption 3.1 holds, and $s_0 = $ DD. Then, for $\alpha \leq \frac{C \log(1/\epsilon)}{\log(1/\delta)}$, where $C$ is a constant that is independent of the discount factor $\gamma$, the step-size $\alpha$, the rewards and the initialization of the $Q$-values, we have that with probability $1 - \delta$, Algorithm 1 does achieve a cooperative policy: Pavlov or lose-shift policy in $O(1/\alpha)$ iterations.*

Theorem 3.3 states that for an arbitrarily large probability, there exist small enough $\alpha > 0$ and $\epsilon > 0$ such that agents following Algorithm 1 converge from an always defect policy to a Pavlov policy. The full proof of Theorem 3.3 can be found in Appendix D. A proof sketch is provided below.

*Proof sketch.* We give the proof sketch for going from

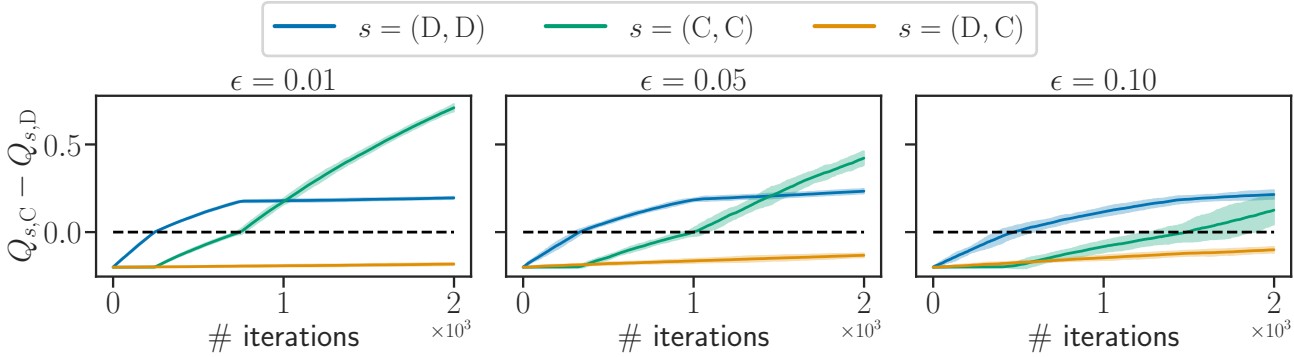

*Figure 3.* **From `always defect` to `Pavlov` policy, Algorithm 1 with exploration** ($g = 1.8$ and $\gamma = 0.6$). Evolution of the $Q$-learning policy as a function of the number of iterations in the iterated prisoner's dilemma. With a correct (optimistic) initialization, players go from an `always defect` policy to the `lose-shift` policy and then to the cooperative `Pavlov` policy.

`always defect` to `lose-shift` with high probability (Step 1). The proof for Theorem 3.3 is three-folded: first, we show that the non-greedy actions are chosen at most $k$ times. This requires controlling the occurring probability of the event

$$\mathcal{E}_{k,T} \triangleq \{\text{either } a_t^1 \text{ or } a_t^2 \text{ is a non-greedy action}$$
$$\text{for } \textit{at most } k \text{ values of } t = 1, \dots, T\} \ ,$$

which is done in Lemma 3.4 *i)*. Using bounds on the $Q$-entries at each iteration (Lemma 3.4 *ii)*), one can control the maximum deviation of the $Q$-entries of non-greedy (Lemma 3.4 *iii)*) and greedy actions (Lemma 3.4 *ii)*). Building on Lemmas 3.4 *ii)* and 3.4 *iii)* one can find assumptions on $k$ and $T$ such that $Q$-learners go from `always defect` to `lose-shift` Lemmas 3.4 *v)* and 3.4 *vi)*.

**Lemma 3.4.** *Let* $0 < \epsilon < 1/2$, $0 < \gamma < 1$ *and* $0 \leq k \leq T$, $k \in \mathbb{N}$. *Suppose that Assumption 3.1 holds,* $s_0 =$ DD, *and both agents are guided by* $\epsilon$*-greedy $Q$-learning (Algorithm 1), then*

i) *The probability of the event* $\mathcal{E}_{k,T}$ *is lower bounded*

$$\mathbb{P}(\mathcal{E}_{k,T}) \geq 1 - 2^T (2\epsilon)^{k+1} \ .$$

ii) *For all state-action pair* $(s, a) \in \mathcal{S} \times \mathcal{A}$

$$|Q_{s,a}^{t+1} - Q_{s,a}^t| \leq \frac{\Delta_r \alpha}{1 - \gamma} \ .$$

iii) *On the event* $\mathcal{E}_{k,T}$, *the deviation for the $Q$-values others than* $Q_{(D,D),D}$ *is at most*

$$|Q_{s,a}^t - Q_{s,a}^{t_0}| \leq \frac{2k\Delta_r \alpha}{1 - \gamma}, \quad \forall (s, a) \neq (DD, D) \ .$$

iv) *On the event* $\mathcal{E}_{k,T}$, *the deviation for the $Q$-value* $Q_{(D,D),D}$ *is upper-bounded*

$$Q_{(D,D),D}^{t+1} - Q_{(D,D),D}^{\star,\text{Defect}} \leq \frac{2k\Delta_r \alpha}{1 - \gamma}$$
$$+ (1 - \alpha(1 - \gamma))^{T - 2k} \left( Q_{(D,D),D}^{t_0} - Q_{(D,D),D}^{\star,\text{Defect}} \right) \ .$$

v) *On the event* $\mathcal{E}_{k,T}$, *for* $k < \frac{(1-\gamma)\Delta Q}{2\alpha\Delta_r}$, *with* $\Delta Q \triangleq \min_{s \neq DD} Q_{s,D}^{t_0} - Q_{s,C}^{t_0}$

$$Q_{s,D}^t > Q_{s,C}^t, \quad \forall t \leq T, \ s \neq \text{DD} \ .$$

vi) *On the event* $\mathcal{E}_{k,T}$, *if* $T > 2k +$
$\frac{\log\left(Q_{(D,D),C}^{t_0} - Q_{(D,D),D}^{\star,\text{Defect}} - \frac{4k\Delta_r \alpha}{1-\gamma}\right) - \log\left(Q_{(D,D),D}^{t_0} - Q_{(D,D),D}^{\star,\text{Defect}}\right)}{\log(1 - \alpha + \gamma\alpha)}$,
*then*

$$Q_{(D,D),D}^T < Q_{(D,D),C}^T \ .$$

Combining Lemmas 3.4 *i)* to 3.4 *vi)* yields that agents learn the `lose-shift` policy with high probability. Similar arguments hold for learning the `Pavlov` policy from the `lose-shift` policy. □

Figure 3 shows the evolution of the $Q$-values as a function of the number of iterations for multiple values of $\epsilon$. 100 of runs are averaged, and the standard deviation is displayed in the shaded area. For $\epsilon = 0.01$, we almost recover the no exploration case. The larger the exploration parameter $\epsilon$, the less the condition for the `Pavlov` policy to be a fixed point of the multi-agent Bellman Equation (4) is satisfied (see Proposition 2.2).

## 4. Related Work

The question of tacit collusion/cooperation between agents has been studied almost independently in the economics

and reinforcement learning literature. On the one hand, the economics community perceives collusion/cooperation as a negative feature since collusion between sellers is against the consumer's interest and remains illegal, violating anti-trust laws. On the other hand, in reinforcement learning, collusion/cooperation is considered a desired property, which would potentially yield symbiotic behaviors between decentralized agents.

**$Q$-learning *without* Memory**. The dynamics of $Q$-learning in the iterated prisoner's dilemma has previously been studied without memory, *i.e.,* Equation (4) with $\mathcal{S} = \emptyset$. More precisely, Banchio & Mantegazza (2022) studied the memoryless, averaged (Seijen et al., 2009; Sutton & Barto, 2018, 6.10), and time-continuous dynamics of Equation (3) (*i.e.,* no stochasticity nor discreteness), with $p(a^1)$ the probability of picking the action $a^1 \in \{C, D\}$ writes

$$\dot{Q}_{a^1} = p(a^1)\left(\mathbb{E}_{a^2 \sim \pi}(r_{a^1, a^2}) + \gamma \max_{a'} Q_{a'} - Q_{a^1}\right) . \quad (8)$$

The main theoretical results from the memoryless case are the following: (i) There is no equilibrium corresponding to a cooperative policy, *i.e.,* no fixed point of Equation (8) such that $Q_D^\star < Q_C^\star$. (ii) There is always an equilibrium corresponding to an `always defect` policy, *i.e.,* a fixed point of Equation (8) such that $Q_D^\star > Q_C^\star$ (iii) Depending on the value of the exploration parameter $\epsilon$ and the incentive to cooperate $g$, an *additional equilibrium at the discontinuity of the vector field* can appear, *i.e.,* an equilibrium where $Q_D^\star = Q_C^\star$. This equilibrium can be seen as partial cooperation and is considered to be an artifact of the memoryless setting.

**Multi-agent Reinforcement Learning**. The problem of cooperation has also been approached in the *multi-agent reinforcement learning* community, which tackles the question of multiple agents' decision-making in a *shared* environment. Depending on the environment and the rewards, agents can either prioritize their interests or promote cooperation. This yielded a vast literature of empirical algorithms aiming at learning cooperative strategies (Whitehead, 1991) in various settings (Lowe et al., 2017; Sunehag et al., 2017; Guan et al., 2023), depending on if the agents can synchronize or not (Arslan & Yüksel, 2016; Yongacoglu et al., 2021; Nekoei et al., 2023; Zhao et al., 2023; Yongacoglu et al., 2023).

Multi-agent reinforcement learning has also been approached from the theoretical side: the convergence of algorithms has also been analyzed in the case of zero-sum games (Littman, 1994), non-zero-sum with only a single Nash equilibrium (Hu & Wellman, 1998). The main difficulty of the multi-agent reinforcement learning theoretical analyses (Zhang et al., 2021, §3) comes from (i) The notion of optimality/learning goal, *i.e.,* what are the desirable properties of the learned policy. (ii) The non-stationary environ-

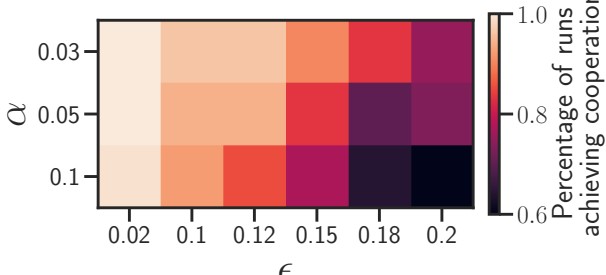

*Figure 4.* **Influence of $\alpha$ and $\epsilon$ on the cooperation.** For each pair $(\alpha, \epsilon)$, the probability of learning a cooperative strategy is estimated with 100 runs. As predicted by Theorem 3.3, cooperation is achieved with a high probability for smaller values of the exploration parameter $\epsilon$, and of the step-size $\alpha$.

ment, which yields rewards depending on the other player's actions. (iii) The existence of multiple equilibria. While the question of convergence towards *a Nash* has previously been studied (Hu & Wellman, 1998; Wainwright, 2019; Usui & Ueda, 2021; Meylahn & Janssen, 2022), characterizing towards *which equilibrium* algorithms converge is much harder, and is usually only studied numerically, via simulations or by proposing approximation methods to reduce the problem to solving a (prefereably smooth) dynamical system which is then analyzed either theoretically or numerically (Kaisers, 2012; Gupta et al., 2017; Barfuss & Meylahn, 2023; Meylahn, 2023; Ding et al., 2023; Cartea et al., 2022).

## 5. Experiments

**Experimental Setup**. In all the experiments we consider a prisoner's dilemma with a fixed incentive to cooperate $g$ and a fixed discount factor $\gamma$: $g = 1.8$ and $\gamma = 0.6$. In Figures 1 to 3 the stepsize $\alpha$ is fixed to $\alpha = .1$

**Influence of Step Size and Exploration**. Figure 4 shows the percentage of runs that achieve cooperation for multiple values of the exploration parameter $\epsilon$ and the step-size $\alpha$. For each pair $(\epsilon, \alpha)$, Algorithm 1 is run 100 times, for 2000 iterations, with an optimistic initialization. The percentage of runs that yields cooperation is displayed as a function of $\epsilon$ and $\alpha$. As predicted by Proposition 2.2, when $\epsilon$ is too large ($\epsilon = 0.2$), `Pavlov` policy is no longer a stable point of Equation (4), and no cooperation is achieved. As predicted by Theorem 3.3, smaller values of $\alpha$ and $\epsilon$ yield a larger probability of cooperation. The influence of the incentive to cooperate $g$ is explored in Appendix E.1: similar behavior are observed for multiple values of the influence to cooperate $g$.

**Similar Behaviour in Deep $Q$-learning.** We train a deep $Q$-network agent (Mnih et al., 2015) with a $Q$-function

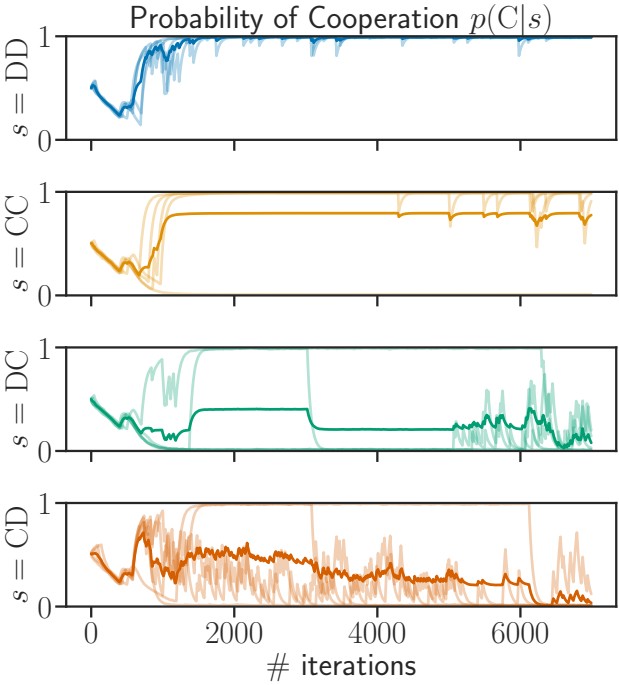

*Figure 5.* **Extension to deep $Q$-learning.** Evolution of the probability to cooperate conditioned on the state, *i.e.,* the previous actions, as a function of the number of iterations in the iterated prisoner's dilemma. Multiple runs corresponding to multiple seeds are displayed, as well as their mean. Agents are trained against a random agent for the initialization and go from an `always defect` policy to the cooperative `Pavlov` policy.

## 6. Conclusion and Limitation

To understand the empirical collusion phenomenon of machine learning algorithms observed in the economics community (Calvano et al., 2020b), we theoretically studied multi-agent $Q$-learning in the minimalist setting of the iterated prisoner's dilemma. We showed that two agents guided by the usual $Q$-learning Algorithm 1 could learn the cooperative `Pavlov` policy, even when both agents start from an `always defect` policy. In addition, we provided explicit conditions (Assumption 3.1) for convergence towards such a cooperative strategy in the fully greedy case ($\epsilon = 0$, Section 3.1), and in the $\epsilon$-greedy case ($\epsilon > 0$, Section 3.2).

The major limitation of our analysis is the self-play assumption, which is currently crucial in the proof of Theorems 3.2 and 3.3. Empirically, *relaxing the self-play assumption still yields cooperation* "most of the time" (Barfuss & Meylahn, 2023, Fig. 1a): in this setting, players alternate between multiple strategies but on average mostly play the `Pavlov` strategy. Hence, relaxing the self-play assumption would require analyzing the stationary distribution of the process (as done in Xu & Zhao 2024 in the memoryless case), which is significantly harder and left as future work. Finally, we also plan to study the extension of the emergence of collusion in other reinforcement learning algorithms and more complex environments, such as policy gradient or Bertrand games.

parameterized by a neural network with two linear layers and ReLu activation function to play the iterated prisoner's dilemma. First, the network is initialized by playing against a random agent for 500 iterations. As discussed in the previous paragraph, this corresponds to Algorithm 2 with an exploration parameter of $\epsilon = 1/2$. Then the agent is playing against itself, with a decreasing exploration going from $\epsilon = 1/2$ to $\epsilon = 10^{-2}$. Details and hyperparameters can be found in Appendix E.2.

**Comments on Figure 5.** A batch of actions is drawn at each iteration, and we compute the empirical probability of choosing the 'cooperate' action given a specific state $p(\mathrm{C}|s)$, for all the previous possible states, $s \in \{\mathrm{CC}, \mathrm{CD}, \mathrm{DC}, \mathrm{DD}\}$. The procedure is repeated for multiple seeds and the mean across the seed is displayed as a thick line. Figure 5 displays the probability of cooperation as a function of the number of iterations. Players start to cooperate in the CD states, then successively in the DD and CC states. Finally, around iteration 6000 players start to defect in the CD state and reach the `Pavlov` policy.

## Acknowledgements

Q.B. would like to thank Samsung Electronics Co., Ldt. for partially funding this research. GG is partially funded by a Canada CIFAR AI chair and and NSERC discovery grant. E.C. acknowledges funding by the European Union (ERC grant AI-Comp, 101098332) and PRIN 2022 grant 'Algorithms and economic choices', codice Cineca 2022S5RC7R, CUP E53D23006420001. This research project was initiated during the research program "Learning and Games" semester at the Simons Institute for the Theory of Computing.

## Impact Statement

This paper presents work whose goal is mainly theoretical advances in the field of machine learning. The findings stress the importance of proactive ethical considerations and regulatory frameworks for responsible machine learning deployment, influencing policy-making and industry practices to ensure fair competition and societal benefit.

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

# A. Proofs of $Q$-values at Convergence

## A.1. `Always defect` Policy

**Proposition 2.1** (Always Defect). *The following Q-table $Q^{\star,\text{Defect}}$ is a fixed point of the Bellman Equation (4) and yields an* `always defect` *policy, $\forall s \in \{\text{C}, \text{D}\}^2$,*

$$Q_{s,\text{D}}^{\star,\text{Defect}} = \mathbb{E}_{a^2 \sim \pi(\cdot|s)} r_{\text{D},a^2}/(1-\gamma) \ ,$$

$$Q_{s,\text{C}}^{\star,\text{Defect}} = Q_{s,\text{D}}^{\star,\text{Defect}} - \mathbb{E}_{a^2 \sim \pi(\cdot|s)} \left( r_{\text{D},a^2} - r_{\text{C},a^2} \right) \ .$$

*Proof of Proposition 2.1 (`always defect`).* The greedy action of the `always defect` policy is to defect all the time. For all states $s_t \in \{\text{C}, \text{D}\}^2$, $a_t^1 \in \{\text{C}, \text{D}\}$, the self-play multi-agent Bellman equation writes

$$Q_{s_t, a_t^1}^{\star,\text{Defect}} = \gamma(1-\epsilon)Q_{(a_t^1,\text{D}),\text{D}}^{\star,\text{Defect}} + \gamma\epsilon Q_{(a_t^1,\text{C}),\text{D}}^{\star,\text{Defect}} + \mathbb{E}_a r_{a_t^1,a} \ , \tag{9}$$

*i.e., , for all state $s_t \in \{\text{C}, \text{D}\}^2$*

$$Q_{s_t,\text{D}}^{\star,\text{Defect}} - \gamma(1-\epsilon)Q_{(\text{DD}),\text{D}}^{\star,\text{Defect}} - \gamma\epsilon Q_{(\text{DC}),\text{D}}^{\star,\text{Defect}} = \mathbb{E}_a r_{\text{D},a} \tag{10}$$

$$Q_{s_t,\text{C}}^{\star,\text{Defect}} - \gamma(1-\epsilon)Q_{(\text{C},\text{D}),\text{D}}^{\star,\text{Defect}} - \gamma\epsilon Q_{(\text{C},\text{C}),\text{D}}^{\star,\text{Defect}} = \mathbb{E}_a r_{\text{C},a} \ . \tag{11}$$

Evaluating Equation (10) in $s_t = (\text{D}, \text{C})$ and $s_t = (\text{DD})$ yields

$$(1 - \gamma\epsilon)Q_{\text{DC},\text{D}}^{\star,\text{Defect}} - \gamma(1-\epsilon)Q_{(\text{D},\text{D}),\text{D}}^{\star,\text{Defect}} = \mathbb{E}_a r_{\text{D},a} \tag{12}$$

$$-\gamma\epsilon Q_{\text{DC},\text{D}}^{\star,\text{Defect}} + (1 - \gamma(1-\epsilon))Q_{\text{DD},\text{D}}^{\star,\text{Defect}} = \mathbb{E}_a r_{\text{D},a} \ . \tag{13}$$

One can see that

$$Q_{\text{DC},\text{D}}^{\star,\text{Defect}} = Q_{(\text{D},\text{D}),\text{D}}^{\star,\text{Defect}} = \mathbb{E}_a r_{\text{D},a}/(1-\gamma) \ , \tag{14}$$

is a solution to the linear system. Plugging Equation (14) into Equation (10) yields

$$Q_{\text{CC},\text{D}}^{\star,\text{Defect}} = Q_{\text{CD},\text{D}}^{\star,\text{Defect}} = \mathbb{E}_a r_{\text{D},a}/(1-\gamma) \ . \tag{15}$$

Plugging Equation (15) into Equation (11) yields

$$\forall s \in \{\text{C}, \text{D}\}^2, \ Q_{s,\text{C}}^{\star,\text{Defect}} = \mathbb{E}_a r_{\text{C},a} + \frac{\gamma}{1-\gamma}\mathbb{E}_a r_{\text{D},a} \tag{16}$$

$$= \frac{1}{1-\gamma}\mathbb{E}_a r_{\text{D},a} - (\mathbb{E}_a r_{\text{D},a} - \mathbb{E}_a r_{\text{C},a}) \tag{17}$$

$$= Q_{s,\text{D}}^{\star,\text{Defect}} - (\mathbb{E}_a r_{\text{D},a} - \mathbb{E}_a r_{\text{C},a}) \ . \tag{18}$$

This means that these $Q$ values imply a greedy `always defect` policy if and only if

$$\mathbb{E}_a r_{\text{D},a} > \mathbb{E}_a r_{\text{C},a} \ , i.e., \tag{19}$$

$$(1-\epsilon)r_{\text{DD}} + \epsilon r_{\text{DC}} > (1-\epsilon)r_{\text{CD}} + \epsilon r_{\text{CC}} \ , \tag{20}$$

which is always true since $r_{\text{DD}} > r_{\text{CD}}$ and $r_{\text{DC}} > r_{\text{CC}}$. $\qquad\square$

## A.2. $Q$-values at convergence for the `Pavlov` Policy

**Proposition 2.2** (Pavlov). *If $\gamma > \frac{r_{\text{DC}} - r_{\text{CC}}}{r_{\text{CC}} - r_{\text{DD}}}$ and $\epsilon$ is small enough, then there exists a Q-function, $Q^{\star,\text{Pavlov}}$, which is a fixed point of the self-play multi-agent Bellman Equation (4) and yields the Pavlov policy, i.e,*

$$\forall s \in \{\text{CC}, \text{DD}\} \quad Q_{s,\text{C}}^{\star,\text{Pavlov}} > Q_{s,\text{D}}^{\star,\text{Pavlov}} \quad and$$

$$\forall s \in \{\text{CD}, \text{DC}\} \quad Q_{s,\text{C}}^{\star,\text{Pavlov}} < Q_{s,\text{D}}^{\star,\text{Pavlov}} \ .$$

*Proof of Q-values at convergence for the* `Pavlov` *Policy.*

$$Q^{\star,\text{Pavlov}}_{s_t,a_t^1} = \gamma(1-\epsilon)Q^{\star,\text{Pavlov}}_{(a_t^1,\bar{a}^2(s_t)),\bar{a}^1(a_t^1,\bar{a}^2(s_t))} + \gamma\epsilon Q^{\star,\text{Pavlov}}_{(a_t^1,\underline{a}^2(s_t)),\bar{a}^1(a_t^1,\underline{a}^2(s_t))} + \mathbb{E}_{a\sim\pi(\cdot|s_t)}\ r_{a_t^1,a} \tag{21}$$

$$Q^{\star,\text{Pavlov}}_{\text{CC},a_t^1} = \gamma(1-\epsilon)Q^{\star,\text{Pavlov}}_{(a_t^1,C),\bar{a}^1(a_t^1,C)} + \gamma\epsilon Q^{\star,\text{Pavlov}}_{(a_t^1,D,,\bar{a}^1(a_t^1,D)} + \mathbb{E}_{a\sim\pi(\cdot|CC)}\ r_{a_t^1,a} \tag{22}$$

$$Q^{\star,\text{Pavlov}}_{\text{DD},a_t^1} = \gamma(1-\epsilon)Q^{\star,\text{Pavlov}}_{(a_t^1,C),\bar{a}^1(a_t^1,C)} + \gamma\epsilon Q^{\star,\text{Pavlov}}_{(a_t^1,D),\bar{a}^1(a_t^1,D)} + \mathbb{E}_{a\sim\pi(\cdot|DD)}\ r_{a_t^1,a} \tag{23}$$

$$Q^{\star,\text{Pavlov}}_{\text{CD},a_t^1} = \gamma(1-\epsilon)Q^{\star,\text{Pavlov}}_{(a_t^1,D),\bar{a}^1(a_t^1,D)} + \gamma\epsilon Q^{\star,\text{Pavlov}}_{(a_t^1,C),\bar{a}^1(a_t^1,C)} + \mathbb{E}_{a\sim\pi(\cdot|CD)}\ r_{a_t^1,a} \tag{24}$$

$$Q^{\star,\text{Pavlov}}_{\text{DC},a_t^1} = \gamma(1-\epsilon)Q^{\star,\text{Pavlov}}_{(a_t^1,D),\bar{a}^1(a_t^1,D)} + \gamma\epsilon Q^{\star,\text{Pavlov}}_{(a_t^1,C),\bar{a}^1(a_t^1,C)} + \mathbb{E}_{a\sim\pi(\cdot|DC)}r_{a_t^1,a} \tag{25}$$

Hence

$$Q^{\star,\text{Pavlov}}_{(C,C),C} - \gamma(1-\epsilon)Q^{\star,\text{Pavlov}}_{(C,C),C} - \gamma\epsilon Q^{\star,\text{Pavlov}}_{(D,D),D} = \mathbb{E}_{a\sim\pi(\cdot|CC)}\ r_{C,a} \tag{26}$$

$$Q^{\star,\text{Pavlov}}_{(C,C),D} - \gamma(1-\epsilon)Q^{\star,\text{Pavlov}}_{(D,C),D} - \gamma\epsilon Q^{\star,\text{Pavlov}}_{(D,D),C} = \mathbb{E}_{a\sim\pi(\cdot|CC)}\ r_{D,a} \tag{27}$$

$$Q^{\star,\text{Pavlov}}_{(D,D),C} - \gamma(1-\epsilon)Q^{\star,\text{Pavlov}}_{(C,C),C} - \gamma\epsilon Q^{\star,\text{Pavlov}}_{(C,D),D} = \mathbb{E}_{a\sim\pi(\cdot|DD)}\ r_{C,a} \tag{28}$$

$$Q^{\star,\text{Pavlov}}_{(D,D),D} - \gamma(1-\epsilon)Q^{\star,\text{Pavlov}}_{(D,C),D} - \gamma\epsilon Q^{\star,\text{Pavlov}}_{(D,D),C} = \mathbb{E}_{a\sim\pi(\cdot|DD)}\ r_{D,a} \tag{29}$$

$$Q^{\star,\text{Pavlov}}_{(C,D),C} - \gamma(1-\epsilon)Q^{\star,\text{Pavlov}}_{(C,D),D} - \gamma\epsilon Q^{\star,\text{Pavlov}}_{(C,C),C} = \mathbb{E}_{a\sim\pi(\cdot|CD)}\ r_{C,a} \tag{30}$$

$$Q^{\star,\text{Pavlov}}_{(C,D),D} - \gamma(1-\epsilon)Q^{\star,\text{Pavlov}}_{(D,D),C} - \gamma\epsilon Q^{\star,\text{Pavlov}}_{(D,C),D} = \mathbb{E}_{a\sim\pi(\cdot|CD)}\ r_{D,a} \tag{31}$$

$$Q^{\star,\text{Pavlov}}_{(D,C),C} - \gamma(1-\epsilon)Q^{\star,\text{Pavlov}}_{(C,D),D} - \gamma\epsilon Q^{\star,\text{Pavlov}}_{(C,C),C} = \mathbb{E}_{a\sim\pi(\cdot|DC)}\ r_{C,a} \tag{32}$$

$$Q^{\star,\text{Pavlov}}_{(D,C),D} - \gamma(1-\epsilon)Q^{\star,\text{Pavlov}}_{(D,D),C} - \gamma\epsilon Q^{\star,\text{Pavlov}}_{(D,C),D} = \mathbb{E}_{a\sim\pi(\cdot|DC)}\ r_{D,a}\ . \tag{33}$$

One can observe that

$$Q^{\star,\text{Pavlov}}_{(C,C),C} = Q^{\star,\text{Pavlov}}_{(D,D),C} \tag{34}$$

$$Q^{\star,\text{Pavlov}}_{(C,C),D} = Q^{\star,\text{Pavlov}}_{(D,D),D} \tag{35}$$

$$Q^{\star,\text{Pavlov}}_{(C,D),C} = Q^{\star,\text{Pavlov}}_{(D,C),C} \tag{36}$$

$$Q^{\star,\text{Pavlov}}_{(C,D),D} = Q^{\star,\text{Pavlov}}_{(D,C),D}\ , \tag{37}$$

and the system of equations rewrites

$$Q^{\star,\text{Pavlov}}_{(C,C),C} - \gamma(1-\epsilon)Q^{\star,\text{Pavlov}}_{(C,C),C} - \gamma\epsilon Q^{\star,\text{Pavlov}}_{(C,D),D} = \mathbb{E}_{a\sim\pi(\cdot|CC)}\ r_{C,a} \tag{38}$$

$$Q^{\star,\text{Pavlov}}_{(D,D),D} - \gamma(1-\epsilon)Q^{\star,\text{Pavlov}}_{(C,D),D} - \gamma\epsilon Q^{\star,\text{Pavlov}}_{(C,C),C} = \mathbb{E}_{a\sim\pi(\cdot|DD)}\ r_{D,a} \tag{39}$$

$$Q^{\star,\text{Pavlov}}_{(C,D),C} - \gamma(1-\epsilon)Q^{\star,\text{Pavlov}}_{(C,D),D} - \gamma\epsilon Q^{\star,\text{Pavlov}}_{(C,C),C} = \mathbb{E}_{a\sim\pi(\cdot|CD)}\ r_{C,a} \tag{40}$$

$$Q^{\star,\text{Pavlov}}_{(C,D),D} - \gamma(1-\epsilon)Q^{\star,\text{Pavlov}}_{(C,C),C} - \gamma\epsilon Q^{\star,\text{Pavlov}}_{(C,D),D} = \mathbb{E}_{a\sim\pi(\cdot|CD)}\ r_{D,a} \tag{41}$$

For $Q^{\star,\text{Pavlov}}_{(C,C),C}$ and $Q^{\star,\text{Pavlov}}_{(C,D),D}$ on needs to solve the following linear system:

$$\begin{pmatrix} 1-\gamma(1-\epsilon) & -\gamma\epsilon \\ -\gamma(1-\epsilon) & 1-\gamma\epsilon \end{pmatrix} \begin{pmatrix} Q^{\star,\text{Pavlov}}_{(C,C),C} \\ Q^{\star,\text{Pavlov}}_{(C,D),D} \end{pmatrix} = \begin{pmatrix} \mathbb{E}_{a\sim\pi(\cdot|CC)}\ r_{C,a} \\ \mathbb{E}_{a\sim\pi(\cdot|CD)}\ r_{D,a} \end{pmatrix}\ , \tag{42}$$

which yields

$$Q^{\star,\text{Pavlov}}_{(D,D),C} = Q^{\star,\text{Pavlov}}_{(C,C),C} = \frac{(1-\gamma\epsilon)\mathbb{E}_{a\sim\pi(\cdot|CC)}\ r_{C,a} + \gamma\epsilon\mathbb{E}_{a\sim\pi(\cdot|CD)}\ r_{D,a}}{1-\gamma} \tag{43}$$

$$Q^{\star,\text{Pavlov}}_{(D,C),D} = Q^{\star,\text{Pavlov}}_{(C,D),D} = \frac{\gamma(1-\epsilon)\mathbb{E}_{a\sim\pi(\cdot|CC)}\ r_{C,a} + (1-\gamma(1-\epsilon))\mathbb{E}_{a\sim\pi(\cdot|CD)}\ r_{D,a}}{1-\gamma}\ . \tag{44}$$

$$Q^{\star,\text{Pavlov}}_{(\text{C},\text{D}),\text{D}} - Q^{\star,\text{Pavlov}}_{(\text{C},\text{D}),\text{C}} = Q^{\star,\text{Pavlov}}_{(\text{D},\text{C}),\text{D}} - Q^{\star,\text{Pavlov}}_{(\text{D},\text{C}),\text{C}} \tag{45}$$

$$= \gamma(1-\epsilon)Q^{\star,\text{Pavlov}}_{(\text{C},\text{C}),\text{C}} + \gamma\epsilon Q^{\star,\text{Pavlov}}_{(\text{C},\text{D}),\text{D}} + \mathbb{E}_{a\sim\pi(\cdot|\text{CD})}\, r_{\text{D},a} \tag{46}$$

$$- \gamma(1-\epsilon)Q^{\star,\text{Pavlov}}_{(\text{C},\text{D}),\text{D}} - \gamma\epsilon Q^{\star,\text{Pavlov}}_{(\text{C},\text{C}),\text{C}} - \mathbb{E}_{a\sim\pi(\cdot|\text{CD})}\, r_{\text{C},a} \tag{47}$$

$$= \gamma(1-2\epsilon)Q^{\star,\text{Pavlov}}_{(\text{C},\text{C}),\text{C}} - \gamma(1-2\epsilon)Q^{\star,\text{Pavlov}}_{(\text{C},\text{D}),\text{D}} + \mathbb{E}_{a\sim\pi(\cdot|\text{CD})}\, r_{\text{D},a} - \mathbb{E}_{a\sim\pi(\cdot|\text{CD})}\, r_{\text{C},a} \tag{48}$$

$$= \gamma(1-2\epsilon)(Q^{\star,\text{Pavlov}}_{(\text{C},\text{C}),\text{C}} - Q^{\star,\text{Pavlov}}_{(\text{C},\text{D}),\text{D}}) + \mathbb{E}_{a\sim\pi(\cdot|\text{CD})}\, r_{\text{D},a} - \mathbb{E}_{a\sim\pi(\cdot|\text{CD})}\, r_{\text{C},a} \tag{49}$$

$$= \gamma(1-2\epsilon)\underbrace{(\mathbb{E}_{a\sim\pi(\cdot|\text{CC})}\, r_{\text{C},a} - \mathbb{E}_{a\sim\pi(\cdot|\text{CD})}\, r_{\text{D},a})}_{\text{depends on }\epsilon} + \underbrace{\mathbb{E}_{a\sim\pi(\cdot|\text{CD})}\, r_{\text{D},a} - \mathbb{E}_{a\sim\pi(\cdot|\text{CD})}\, r_{\text{C},a}}_{>0} \tag{50}$$

$$= \gamma(1-2\epsilon)\underbrace{((1-\epsilon)(r_{\text{CC}}-r_{\text{DD}}) + \epsilon(r_{\text{CD}}-r_{\text{DC}}))}_{\text{depends on }\epsilon} + \underbrace{\mathbb{E}_{a\sim\pi(\cdot|\text{CD})}\, r_{\text{D},a} - \mathbb{E}_{a\sim\pi(\cdot|\text{CD})}\, r_{\text{C},a}}_{>0} \tag{51}$$

$$= \gamma(1-2\epsilon)(2(g-1)-2g\epsilon) + (2-g)\,, \tag{52}$$

Which is positive. In addition

$$Q^{\star,\text{Pavlov}}_{(\text{D},\text{D}),\text{C}} - Q^{\star,\text{Pavlov}}_{(\text{D},\text{D}),\text{D}} \tag{53}$$

$$= \gamma(1-2\epsilon)\underbrace{(\mathbb{E}_{a\sim\pi(\cdot|\text{CC})}\, r_{\text{C},a} - \mathbb{E}_{a\sim\pi(\cdot|\text{CD})}\, r_{\text{D},a})}_{\text{depends}} + \underbrace{\mathbb{E}_{a\sim\pi(\cdot|\text{DD})}\, (r_{\text{C},a} - r_{\text{D},a})}_{<0} \tag{54}$$

$$= \gamma(1-2\epsilon)\underbrace{((1-\epsilon)(r_{\text{CC}}-r_{\text{DD}}) + \epsilon(r_{\text{CD}}-r_{\text{DC}}))}_{\text{depends on }\epsilon} + \underbrace{(1-\epsilon)(r_{\text{CC}}-r_{\text{DC}}) + \epsilon(r_{\text{CD}}-r_{\text{DD}})}_{<0} \tag{55}$$

$$= 2\gamma(1-2\epsilon)\underbrace{(((g-1)-g\epsilon))}_{\text{depends on }\epsilon} + g\,, \tag{56}$$

which is positive for $\epsilon$ small enough.

$\square$

## B. Propositions B.1 and B.2

**Proposition B.1** (`Always defect`). *For all $\gamma \in (0,1)$, the `always defect` policy, i.e., $\pi^{\text{Defect}}(\text{D}|s) = 1\,, \forall s \in \mathcal{S}$, is a subgame perfect equilibrium.*

*Proof.* Regardless of the state, given that the opponent is always defecting, the best response at each time step is to defect since $r_1(\text{C},\text{D}) < r_1(\text{D},\text{D})$. $\square$

**Proposition B.2** (`Pavlov`). *If $1 > \gamma \geq \frac{2-g}{2(g-1)}$, then the `Pavlov` policy, i.e., $\pi^{\text{Pavlov}}(\text{D}|s) = 1\,, \forall s \in \{\text{CD}, \text{DC}\}$ and $\pi^{\text{Pavlov}}(\text{C}|s) = 1\,, \forall s \in \{\text{DD}, \text{CC}\}$, is a subgame perfect equilibrium.*

*Proof.* Let us assume that our opponent plays according to `Pavlov` and show that the best response is to also adopt this strategy. By the one deviation principle (Osborne, 2004, Prop 438.1), one only needs to show that for the four possible starting states the best response against `Pavlov` is to follow `Pavlov`.

• If we start in $s_0 \in \{\text{CD}, \text{DC}\}$, the opponent will defect and the best response is to defect since, if we cooperate, we end up in $s_1 = \text{CD}$ which leads to a worse payoff that going straight to DD by defecting, $r_{\text{CD}} + \gamma r_{\text{DD}} + \gamma^2 r_{\text{CC}} \leq r_{\text{DD}} + \gamma r_{\text{CC}} + \gamma^2 r_{\text{CC}}$, which is always the case since $r_{\text{CD}} < r_{\text{DD}} < r_{\text{CC}}$.

• If we start in $s_0 = \text{CC}$, we should show that it is better to cooperate than to defect. It is the case if, $r_{\text{CD}} + \gamma r_{\text{DD}} + \gamma^2 r_{\text{CC}} \leq r_{\text{CC}} + \gamma r_{\text{CC}} + \gamma^2 r_{\text{CC}}$, which is true if and only if $\gamma \geq \frac{r_{\text{DC}}-r_{\text{CC}}}{r_{\text{CC}}-r_{\text{DD}}} = \frac{2-g}{2(g-1)}$.

• Finally, if we start in $s_0 = \text{DD}$, it is better to cooperate than to defect as the opponent is cooperating. Hence, the best response to the `Pavlov` policy is to play according to `Pavlov`. $\square$

# C. Proof the Fully Greedy Case (Theorem 3.2)

## C.1. `Lose-shift` to `Pavlov`

Equations (6) and (7) that govern the dynamics of $Q_{(C,C),D}$ and $Q_{(D,D),C}$ from the `Lose-shift` policy to `Pavlov` policy are recalled below:

$$Q_{(C,C),D}^{t+1} = (1-\alpha)Q_{(C,C),D}^t + \alpha\left(r_{DD} + \gamma Q_{(D,D),C}^t\right) \; , \tag{6}$$

$$Q_{(D,D),C}^{t+1} = (1-\alpha)Q_{(D,D),C}^t + \alpha\left(r_{CC} + \gamma Q_{(C,C),D}^{t+1}\right) \; . \tag{7}$$

Plugging Equation (6) in Equation (7) yields:

$$Q_{(D,D),C}^{t+1} = (1-\alpha)Q_{(D,D),C}^t + \alpha\left(r_{CC} + \gamma(1-\alpha)Q_{(C,C),D}^t + \gamma\alpha\left(r_{DD} + \gamma Q_{(D,D),C}^t\right)\right) \; , \textit{i.e.,} \tag{57}$$

$$Q_{(D,D),C}^{t+1} = \alpha\gamma(1-\alpha)Q_{(C,C),D}^t + (1-\alpha+\alpha\gamma^2)Q_{(D,D),C}^t + \alpha\left(r_{CC} + \gamma\alpha r_{DD}\right) \; , \tag{58}$$

Following Equations (6) and (58), the dynamics of $Q_{(C,C),D}$ and $Q_{(D,D),C}$ is governed by the linear system

$$\begin{pmatrix} Q_{(C,C),D}^{t+1} \\ -Q_{(D,D),C}^{t+1} \end{pmatrix} = \begin{pmatrix} 1-\alpha & -\alpha\gamma \\ -\alpha\gamma(1-\alpha) & 1-\alpha+(\alpha\gamma)^2 \end{pmatrix} \begin{pmatrix} Q_{(C,C),D}^t \\ -Q_{(D,D),C}^t \end{pmatrix} + b \; , \tag{59}$$

for a given $b$. Let $(-Q_{(C,C),D}^{\star,\text{lose-shift}}, Q_{(D,D),C}^{\star,\text{lose-shift}})$ be the fixed point of this linear system, it writes

$$\begin{pmatrix} Q_{(C,C),D}^{t+1} - Q_{(C,C),D}^{\star,\text{lose-shift}} \\ -(Q_{(D,D),C}^{t+1} - Q_{(D,D),C}^{\star,\text{lose-shift}}) \end{pmatrix} = \begin{pmatrix} 1-\alpha & -\alpha\gamma \\ -\alpha\gamma(1-\alpha) & 1-\alpha+(\alpha\gamma)^2 \end{pmatrix} \begin{pmatrix} Q_{(C,C),D}^t - Q_{(C,C),D}^{\star,\text{lose-shift}} \\ -(Q_{(D,D),C}^t - Q_{(D,D),C}^{\star,\text{lose-shift}}) \end{pmatrix} \; , \tag{60}$$

Standard computations yield the eigenvalues of the matrix

$$\begin{pmatrix} 1-\alpha & -\alpha\gamma \\ -\alpha\gamma(1-\alpha) & 1-\alpha+(\alpha\gamma)^2 \end{pmatrix} \; . \tag{61}$$

are

$$\lambda_\pm = 1 - \alpha + \alpha\gamma\left(\frac{\alpha\gamma}{2} \pm \sqrt{1-\alpha+\frac{(\alpha\gamma)^2}{4}}\right) \; . \tag{62}$$

Thus for $0 < \alpha < 1$ and $0 < \gamma < 1$ we have $0 < \lambda_\pm < 1$ and thus $\left(Q_{(C,C),D}^t, Q_{(D,D),C}^t\right)$ converges linearly towards $\left(Q_{(C,C),D}^{\star,\text{lose-shift}}, Q_{(D,D),C}^{\star,\text{lose-shift}}\right)$.

Then using Lemma C.1, we have that $Q_{(C,C),D}^t$ will get below $Q_{(C,C),C}^{t_0}$ before $Q_{(D,D),C}^t$ getting below $Q_{(D,D),D}^{t_0}$.

**Lemma C.1.** *In the phase from* `lose-shift` *to* `Pavlov`, *if* $\frac{r_{DD}+\gamma r_{CC}}{1-\gamma^2} \triangleq Q_{(C,C),C}^{\star,\text{lose-shift}} < Q_{(C,C),C}^{t_0} \leq \frac{r_{CC}}{1-\gamma}$, *then while* $Q_{(C,C),D}^t > Q_{(C,C),C}^{t_0}$, $Q_{(D,D),C}^t > Q_{(D,D),D}^{t_0}$.

*Proof.* (Lemma C.1) In phase 2, $Q_{(C,C),D}^{t+1} > Q_{(C,C),C}^{t_0}$ Using Equation (7)

$$Q_{(D,D),C}^{t+1} = (1-\alpha)Q_{(D,D),C}^t + \alpha\big( \qquad r_{CC} + \gamma \underbrace{Q_{(C,C),D}^{t+1}}_{\substack{>Q_{(C,C),C}^{t+1}}} \qquad \big) \; , \tag{63}$$

$$>Q_{(C,C),C}^{t_0}>Q_{(D,D),D}^{t_0}=Q_{(D,D),D}^{t+1}\text{Assumption 3.1 \textit{iii})}$$

hence

$$Q_{(D,D),C}^{t+1} > Q_{(D,D),D}^{t+1} \; . \tag{64}$$

For phases 2 and 3, the convergence is also linear, and similar arguments hold. $\qquad \square$

## C.2. $\mathcal{O}(1/\alpha)$ convergence rate

The convergence of the $Q$-values is linear in each phase (1 to 3). For instance, in phase 1 the convergence is linear with rate $(1 - \alpha(1 - \gamma))$, more specifically,

$$Q^t_{(D,D),D} - Q^{\star,\text{Defect}}_{(D,D),D} = (1 - \alpha(1 - \gamma))^t (Q^{t_0}_{(D,D),D} - Q^{\star,\text{Defect}}_{(D,D),D}) \quad . \tag{65}$$

With the later decrease, the policy switches, *i.e.,*

$$Q^t_{(D,D),D} - Q^{\star,\text{Defect}}_{(D,D),D} < Q^{t_0}_{(D,D),C} - Q^{\star,\text{Defect}}_{(D,D),D}$$

will require a number of steps

$$T = (\log(Q^{t_0}_{(DD),C} - Q^{\star,\text{Defect}}_{(D,D),D})) - \log(Q^{t_0}_{(D,D),D} - Q^{\star,\text{Defect}}_{(D,D),D})) / \log(1 - \alpha(1 - \gamma)) \tag{66}$$

$$\sim_{\alpha \to 0} (\log(Q^{t_0}_{(D,D),D} - Q^{\star,\text{Defect}}_{(D,D),D})) - \log(Q^{t_0}_{(DD),C} - Q^{\star,\text{Defect}}_{(D,D),D}))) / \alpha(1 - \gamma) \tag{67}$$

$$= \mathcal{O}(1/\alpha) \quad . \tag{68}$$

For phases 2 and 3, the convergence is also linear, and similar arguments hold. To summarize, the convergence is linear in the three phases (from `always defect` to `Lose-shift`, from `Lose-shift` to `Pavlov`, staying in `Pavlov`), and is done in $\mathcal{O}(1/\alpha)$ in each phase.

# D. Proof of the $\epsilon$-Greedy Case (Theorem 3.3)

Let us start with the proof of Lemma 3.4.

### D.1. Proof of Lemma 3.4: from `always defect` to `lose-shift`

**Lemma 3.4.** *Let $0 < \epsilon < 1/2$, $0 < \gamma < 1$ and $0 \leq k \leq T$, $k \in \mathbb{N}$. Suppose that Assumption 3.1 holds, $s_0 = DD$, and both agents are guided by $\epsilon$-greedy $Q$-learning (Algorithm 1), then*

i) *The probability of the event $\mathcal{E}_{k,T}$ is lower bounded*

$$\mathbb{P}(\mathcal{E}_{k,T}) \geq 1 - 2^T (2\epsilon)^{k+1} \quad .$$

ii) *For all state-action pair $(s, a) \in \mathcal{S} \times \mathcal{A}$*

$$|Q^{t+1}_{s,a} - Q^t_{s,a}| \leq \frac{\Delta_r \alpha}{1 - \gamma} \quad .$$

iii) *On the event $\mathcal{E}_{k,T}$, the deviation for the $Q$-values others than $Q_{(D,D),D}$ is at most*

$$|Q^t_{s,a} - Q^{t_0}_{s,a}| \leq \frac{2k\Delta_r \alpha}{1 - \gamma}, \quad \forall (s, a) \neq (DD, D) \quad .$$

iv) *On the event $\mathcal{E}_{k,T}$, the deviation for the $Q$-value $Q_{(D,D),D}$ is upper-bounded*

$$Q^{t+1}_{(D,D),D} - Q^{\star,\text{Defect}}_{(D,D),D} \leq \frac{2k\Delta_r \alpha}{1 - \gamma}$$
$$+ (1 - \alpha(1 - \gamma))^{T-2k} \left( Q^{t_0}_{(D,D),D} - Q^{\star,\text{Defect}}_{(D,D),D} \right) \quad .$$

v) *On the event $\mathcal{E}_{k,T}$, for $k < \frac{(1-\gamma)\Delta Q}{2\alpha\Delta_r}$, with $\Delta Q \triangleq \min_{s \neq DD} Q^{t_0}_{s,D} - Q^{t_0}_{s,C}$*

$$Q^t_{s,D} > Q^t_{s,C}, \quad \forall t \leq T, s \neq DD \quad .$$

vi) *On the event $\mathcal{E}_{k,T}$, if $T > 2k + \frac{\log\left( Q^{t_0}_{(D,D),C} - Q^{\star,\text{Defect}}_{(D,D),D} - \frac{4k\Delta_r\alpha}{1-\gamma} \right) - \log\left( Q^{t_0}_{(D,D),D} - Q^{\star,\text{Defect}}_{(D,D),D} \right)}{\log(1 - \alpha + \gamma\alpha)}$, then*

$$Q^T_{(D,D),D} < Q^T_{(D,D),C} \quad .$$

### D.1.1. PROOF OF LEMMA 3.4 *i)*

*Proof of Lemma 3.4* i). The proof of this result relies on the fact that

- The probability of the greedy action is $1 - \epsilon$.

- Each agent picks their action independently of the other.

Thus, either both agents take a greedy action with probability $(1 - \epsilon)^2$, or at least one of them picks a non-greedy action with probability $2\epsilon - \epsilon^2$.

$$\mathbb{P}(\mathcal{E}_{k,T}) = \sum_{i=0}^{k} \mathbb{P}(\text{a non-greedy action has been picked exactly } i \text{ times in } T \text{ steps}) \tag{69}$$

$$= \sum_{i=0}^{k} \binom{T}{i} (1 - \epsilon)^{2(T-i)} (2\epsilon - \epsilon^2)^i \tag{70}$$

$$= 1 - \sum_{i=k+1}^{T} \binom{T}{i} (1 - \epsilon)^{2(T-i)} \overbrace{(2\epsilon - \epsilon^2)^i}^{\leq (2\epsilon)^i \leq (2\epsilon)^{k+1}} \quad (\text{because } 0 \leq \epsilon \leq 1/2) \tag{71}$$

$$\geq 1 - (2\epsilon)^{k+1} \overbrace{\sum_{i=k+1}^{T} \binom{T}{i} (1 - \epsilon)^{2(T-i)}}^{\leq \sum_{i=0}^{T} \binom{T}{i} = 2^T} \tag{72}$$

$$\geq 1 - 2^T (2\epsilon)^{k+1} \; , \tag{73}$$

*i.e.,*

$$\boxed{\mathbb{P}(\mathcal{E}_{k,T}) \geq 1 - 2^T (2\epsilon)^{k+1} \; .} \tag{74}$$

$\square$

We will then use this lemma in steps 1 and 2. The idea of the proof will be to leverage this lower bound to show that the $\epsilon$-greedy dynamic behaves similarly to the fully greedy policy.

### D.1.2. PROOF OF LEMMA 3.4 *ii)*

*Proof of Lemma 3.4* ii). For all state, action pair $(s, a) \in \mathcal{S} \times \mathcal{A}$, the $Q$-learning update rule writes

$$Q_{s,a}^{t+1} = Q_{s,a}^t + \alpha \left( r_t + \gamma \max_{a'} Q_{s,a'}^t - Q_{s,a}^t \right) \; . \tag{75}$$

In addition, since $Q = \sum_{t=0}^{\infty} \gamma^t r_t$, then the $Q$ values cannot go above (resp. below) the maximal (resp. minimal) reward. In other words

$$\frac{r_{\min}}{1 - \gamma} \leq Q_{s,a} \leq \frac{r_{\max}}{1 - \gamma} \quad \text{which yields} \tag{76}$$

$$r_{\min} + \gamma \frac{r_{\min}}{1 - \gamma} - \frac{r_{\max}}{1 - \gamma} \leq r_t + \gamma \max_{a'} Q_{s,a'}^t - Q_{s,a}^t \leq r_{\max} + \gamma \frac{r_{\max}}{1 - \gamma} - \frac{r_{\min}}{1 - \gamma} \tag{77}$$

$$\frac{r_{\min} - r_{\max}}{1 - \gamma} \leq r_t + \gamma \max_{a'} Q_{s,a'}^t - Q_{s,a}^t \leq \frac{r_{\max} - r_{\min}}{1 - \gamma} \; . \tag{78}$$

Plugging Equation (78) in Equation (75) yields the desired result

$$\boxed{|Q_{s,a}^{t+1} - Q_{s,a}^t| \leq \frac{\Delta_r \alpha}{1 - \gamma} \; ,} \tag{79}$$

with $\Delta_r \triangleq r_{\max} - r_{\min}$ the difference between the maximal and the minimal reward. $\square$

### D.1.3. PROOF OF LEMMA 3.4 *iii)*

*Proof of Lemma 3.4* iii). Conditioning on $\mathcal{E}_{k,T}$, for all state-action pair $(s,a) \neq (\mathrm{DD}, \mathrm{D})$, $Q_{s,a}$ is updated at most $2k$ times. Since $Q_{s,a}$ is updated at most $2k$ times, Lemma 3.4 *ii)* repeatedly applied $2k$ times yields

$$|Q_{s,a}^t - Q_{s,a}^{t_0}| \leq \frac{2k\Delta_r \alpha}{1-\gamma}, \quad \forall(s,a) \neq (\mathrm{DD}, \mathrm{D}) \ . \tag{80}$$

$\square$

### D.1.4. PROOF OF LEMMA 3.4 *iv)*

*Proof of Lemma 3.4* iv). Let us consider the first step, from `always defect` to `lose-shift`. We start from the `always defect` region, *i.e.,* the greedy action is to defect regardless of the state. When the greedy action D is picked in the state DD, the entry $Q_{(\mathrm{D},\mathrm{D}),\mathrm{D}}$ is updated as follows. In other words, conditioning on $\mathcal{E}_{k,T}$

- The defect action D in the state DD is played at least $T - 2k$ times. When it is the case, $Q_{(\mathrm{D},\mathrm{D}),\mathrm{D}}$ is updated at least $T - 2k$ times according to

$$Q_{(\mathrm{D},\mathrm{D}),\mathrm{D}}^{t+1} = Q_{(\mathrm{D},\mathrm{D}),\mathrm{D}}^t + \alpha \left( r_{\mathrm{DD}} + \gamma Q_{(\mathrm{D},\mathrm{D}),\mathrm{D}}^t - Q_{(\mathrm{D},\mathrm{D}),\mathrm{D}}^t \right) \tag{81}$$

$$Q_{(\mathrm{D},\mathrm{D}),\mathrm{D}}^{t+1} - Q_{(\mathrm{D},\mathrm{D}),\mathrm{D}}^{\star,\mathrm{Defect}} = (1 - \alpha + \alpha\gamma) \left( Q_{(\mathrm{D},\mathrm{D}),\mathrm{D}}^t - Q_{(\mathrm{D},\mathrm{D}),\mathrm{D}}^{\star,\mathrm{Defect}} \right) \ , \tag{82}$$

with $Q_{(\mathrm{D},\mathrm{D}),\mathrm{D}}^{\star,\mathrm{Defect}} = \frac{r_{\mathrm{DD}}}{1-\gamma}$.

- Otherwise $Q_{(\mathrm{D},\mathrm{D}),\mathrm{D}}$ update is bounded $2k$ times using Lemma 3.4 *ii)*:

$$Q_{(\mathrm{D},\mathrm{D}),\mathrm{D}}^{t+1} - Q_{(\mathrm{D},\mathrm{D}),\mathrm{D}}^{\star,\mathrm{Defect}} \leq Q_{(\mathrm{D},\mathrm{D}),\mathrm{D}}^t - Q_{(\mathrm{D},\mathrm{D}),\mathrm{D}}^{\star,\mathrm{Defect}} + \frac{\Delta_r \alpha}{1-\gamma} \ . \tag{83}$$

Combining Equation (82) for at least $T - 2k$ steps, and Equation (83) for at most $2k$ steps yields

$$Q_{(\mathrm{D},\mathrm{D}),\mathrm{D}}^T - Q_{(\mathrm{D},\mathrm{D}),\mathrm{D}}^{\star,\mathrm{Defect}} \leq (1 - \alpha + \alpha\gamma)^{T-2k} \left( Q_{(\mathrm{D},\mathrm{D}),\mathrm{D}}^{t_0} - Q_{(\mathrm{D},\mathrm{D}),\mathrm{D}}^{\star,\mathrm{Defect}} \right) + \frac{2k\Delta_r \alpha}{1-\gamma} \ . \tag{84}$$

$\square$

### D.1.5. PROOF OF LEMMA 3.4 *v)*

*Proof.* Let $s \in \mathcal{S} \backslash \mathrm{DD}$, $t \leq T$, on the event $\mathcal{E}_{k,T}$

$$Q_{s,\mathrm{D}}^t - Q_{s,\mathrm{C}}^t = \underbrace{Q_{s,\mathrm{D}}^t - Q_{s,\mathrm{D}}^{t_0}}_{\geq -\frac{2k\Delta_r \alpha}{1-\gamma} \text{(using Lemma 3.4 iii))}} + \underbrace{Q_{s,\mathrm{C}}^{t_0} - Q_{s,\mathrm{C}}^t}_{\geq -\frac{2k\Delta_r \alpha}{1-\gamma} \text{(using Lemma 3.4 iii))}} + \underbrace{Q_{s,\mathrm{D}}^{t_0} - Q_{s,\mathrm{C}}^{t_0}}_{\geq \min_{s \neq \mathrm{DD}} Q_{s,\mathrm{D}}^{t_0} - Q_{s,\mathrm{C}}^{t_0} \triangleq \Delta Q^{t_0}} \tag{85}$$

$$\geq -\frac{4k\Delta_r \alpha}{1-\gamma} + \Delta Q^{t_0} \ . \tag{86}$$

Hence, if

$$k > \frac{(1-\gamma)\Delta Q^{t_0}}{4\Delta r} \ , \tag{87}$$

then on the event $\mathcal{E}_{k,T}$ for all $s \in \mathcal{S} \backslash \mathrm{DD}$,

$$Q_{s,\mathrm{D}}^T - Q_{s,\mathrm{C}}^T \geq 0 \ . \tag{88}$$

Note that $\Delta Q^{t_0} \triangleq \min_{s \neq \mathrm{DD}} Q_{s,\mathrm{D}}^{t_0} - Q_{s,\mathrm{C}}^{t_0} > 0$ since one starts from the `always defect` policy. $\square$

### D.1.6. PROOF OF LEMMA 3.4 *vi)*

*Proof.* Using Lemma 3.4 *iv)*, on the event $\mathcal{E}_{k,T}$, one has

$$Q^T_{(D,D),D} \leq (1 - \alpha + \alpha\gamma)^{T-2k} \left( Q^{t_0}_{(D,D),D} - Q^{\star,\text{Defect}}_{(D,D),D} \right) + \frac{2k\Delta_r\alpha}{1-\gamma} + Q^{\star,\text{Defect}}_{(D,D),D} \tag{89}$$

$$\leq -\frac{2k\alpha\Delta_r}{1-\gamma} + Q^{t_0}_{DD,C} \text{ if} \tag{90}$$

$$T > 2k + \frac{\log\left( Q^{t_0}_{(D,D),C} - Q^{\star,\text{Defect}}_{(D,D),D} - \frac{4k\Delta_r\alpha}{1-\gamma} \right) - \log\left( Q^{t_0}_{(D,D),D} - Q^{\star,\text{Defect}}_{(D,D),D} \right)}{\log(1-\alpha+\gamma\alpha)} \ . \tag{91}$$

*Remark* D.1. $\left( Q^{t_0}_{(D,D),C} - Q^{\star,\text{Defect}}_{(D,D),D} - \frac{4k\Delta_r\alpha}{1-\gamma} \right)$ needs to be positive.

In addition, using Lemma 3.4 *iii)* on $\mathcal{E}_{k,T}$ yields

$$-\frac{2k\alpha\Delta_r}{1-\gamma} + Q^{t_0}_{DD,C} \leq Q^T_{DD,C} \ . \tag{92}$$

Combining Equations (90) and (92), yields that on $\mathcal{E}_{k,T}$, if

$$T > 2k + \frac{\log\left( Q^0_{(D,D),C} - Q^{\star,\text{Defect}}_{(D,D),D} - \frac{4k\Delta_r\alpha}{1-\gamma} \right) - \log\left( Q^0_{(D,D),D} - Q^{\star,\text{Defect}}_{(D,D),D} \right)}{\log(1-\alpha+\gamma\alpha)} \ , \tag{93}$$

then

$$Q^T_{(D,D),D} < Q^T_{(D,D),C} \ . \tag{94}$$

$\square$

### D.2. Proof of Theorem 3.3

*Proof of Theorem 3.3.* To conclude, if we set $k = O(1/\alpha)$, such that $k < \frac{\Delta_Q(1-\gamma)}{2\alpha\Delta_r}$ and the total number of steps $T$ such that $T > 2k + \frac{\log\left( Q^{t_0}_{(D,D),C} - Q^{\star,\text{Defect}}_{(D,D),D} - \frac{4k\Delta_r\alpha}{1-\gamma} \right) - \log\left( Q^0_{(D,D),D} - Q^{\star,\text{Defect}}_{(D,D),D} \right)}{\log(1-\alpha+\gamma\alpha)}$, one that with the probability of at least $1 - 2^T \epsilon^{T-k}$, there exists $t_1 < T$ such that $Q^{t_1}_{(D,D),D} < Q^{t_1}_{(D,D),C}$.

The proof can be concluded as follows: in order to hold with probability at least $1 - \delta$, $\epsilon$ must satisfy

$$T\log(2) + k\log 2\epsilon \leq \log\delta \tag{95}$$

$$\log(\epsilon) \leq \underbrace{\frac{1}{k} \log(\delta)}_{:=C/\alpha} - \underbrace{\frac{T}{k} \log(2)}_{:=C_1 \perp\!\!\!\perp \alpha} \ , \tag{96}$$

where the constant $C$ depends on the constants of $T$ and $k$ but is independent of alpha. Indeed, as both $T$ and $k$ grow as $1/\alpha$, the ration $T/k$ can be chosen independent of $\alpha$: this ensures that the event holds with high probability, and does not contradict $k \leq \Delta_Q \frac{(1-\gamma)}{2\alpha\Delta_r} = O(1/\alpha)$ from Theorem 3.3. More formally, the number of steps needed $T$, varies as a function of $1/\alpha$: $T \sim C_1/\alpha$, and $k \sim C_2/\alpha$, for given $C_1, C_2 \in \mathbb{R}$. The choice of $k$ in Theorem 3.3 requires as well a growth as $1/\alpha$: $k \leq \Delta_Q(1-\gamma)/(2\alpha\Delta_r) = \mathcal{O}(1/\alpha)$.

To summarize, we proved that with high probability, there exists a time step $t_1$ such that

$$Q^{t_1}_{(D,D),D} < Q^{t_1}_{(D,D),C} \tag{97}$$

$$Q^{t_1}_{(C,C),D} > Q^{t_1}_{(C,C),C} \tag{98}$$

$$Q^{t_1}_{(D,C),D} > Q^{t_1}_{(D,C),C} \tag{99}$$

$$Q^{t_1}_{(C,D),D} > Q^{t_1}_{(C,D),D} \ , \tag{100}$$

and for all $t < t_1$

$$Q^t_{(D,D),D} > Q^t_{(D,D),C} \tag{101}$$

$$Q^t_{(C,C),D} > Q^t_{(C,C),C} \tag{102}$$

$$Q^t_{(D,C),D} > Q^t_{(D,C),C} \tag{103}$$

$$Q^t_{(C,D),D} > Q^t_{(C,D),D} \ . \tag{104}$$

In plain words, the $Q$-values went from the `always defect` to the `lose-shift` region. □

### D.3. From `lose-shift` to `Pavlov`

In this section, we show similar results to Lemma 3.4. For simplicity, we considered that in Equations (6) and (7) one-time step was performing two $\epsilon$-greedy updates.

1. We can lower-bound $\mathbb{P}(\mathcal{E}_{k,T})$

2. Under the event $\mathcal{E}_{k,T}$ we will observe at least $(1-4k)/2$ pair of updates following Equations (6) and (7) are performed and at most $4k$ greedy updates.

3. Thus the $Q$-values for the greedy actions $Q_{(C,C),D}$ and $Q_{(D,D),C}$ will converge linearly and can be bounded.

4. The $Q$-entries of the non-greedy actions can be bounded.

Similarly to Lemma 3.4, one can control the $Q$-values in the greedy and non-greedy states.

**Lemma D.2.** *Let us consider Algorithm 1 with $\epsilon < 1/2$, $\gamma > 0$ and $\delta > 0$. If both agents act according to an $\epsilon$-greed* `lose-shift` *policy then,*

i) *On the event $\mathcal{E}_{k,T}$, for all $(s,a) \in \mathcal{S} \times \mathcal{A}$*

$$|Q^{t+1}_{s,a} - Q^t_{s,a}| \le \frac{\Delta_r \alpha}{1-\gamma} \ .$$

ii) *On the event $\mathcal{E}_{k,T}$, for all $t \ge t_1$ the deviation for the $Q$-values others than $Q_{(D,D),C}$ and $Q_{(C,C),D}$ is at most*

$$|Q^t_{s,a} - Q^{t_1}_{s,a}| \le \frac{2k\Delta_r \alpha}{1-\gamma}, \quad \forall (s,a) \notin \{(DD,C),(CC,D)\} \ .$$

iii) *On the event $\mathcal{E}_{k,T}$, the deviation for the $Q$-value $Q_{(C,C),D}$ is upper-bounded*

$$Q^t_{(C,C),D} - Q^{\star,lose-shift}_{(C,C),D} \le C_1 \lambda_1^{t-k} + C_2 \lambda_2^{t-k} + \frac{2k\Delta_r \alpha}{1-\gamma} \ . \tag{105}$$

*where $C_1$ and $C_2$ are defined in Equation (123).*

iv) *On the event $\mathcal{E}_{k,T}$, for $k < \frac{(1-\gamma)\Delta Q}{2\alpha\Delta_r}$, with $\Delta Q \triangleq \min_{s \notin \{DD,CC\}} Q^{t_0}_{s,D} - Q^{t_0}_{s,C}$*

$$Q^t_{s,D} > Q^t_{s,C}, \quad \forall t \le T, \ s \ne \{DD, CC\} \ .$$

v) *If $Q^{t_0}_{(C,C),D} - Q^{t_0}_{(C,C),C} \ge \frac{4k\Delta_r \alpha}{1-\gamma}$ then, for all $t_1 \le t \le t_2$*

$$Q^t_{(D,D),C} - Q^t_{(D,D),D} + Q^t_{(C,C),D} - Q^t_{(C,C),C} \ge \frac{k\Delta_r \alpha}{1-\gamma} \ . \tag{106}$$

*In other words, this ensures that the agents either go to the `Pavlov` policy or oscillate between `always defect` and `lose-shift`, but do not directly go from `always defect` to `Pavlov`.*

The proof of Lemmas D.2 *i)* to D.2 *iv)* are the same as for Lemma 3.4.

*Final proof of Theorem 3.3.* Once we reached the `lose-shift` policy, using Lemmas D.2 *iii)* and D.2 *iv)*, either

- $Q^t_{(C,C),D} < Q^t_{(C,C),C}$, and the policy changes for the `Pavlov` policy.

- Either non-greedy actions are taken, and we go to the always `always defect` policy, *i.e.,* we back to the previous situation, Lemma 3.4, but at this time-step, $Q^t_{(C,C),D}$ might be closer to $Q^t_{(C,C),C}$. In this situation
    - Either greedy actions are taken and we achieve again the `lose-shift` region
    - Either non-greedy actions are taken, and $Q^t_{(C,C),D}$ might become smaller than $Q^t_{(C,C),C}$, which yield a `win-stay` policy.

$\square$

*Proof of Lemma D.2 v).*

$$\begin{pmatrix} Q^{t+1}_{(C,C),D} - Q^{\star,\text{lose-shift}}_{(C,C),D} \\ Q^{t+1}_{(D,D),C} - Q^{\star,\text{lose-shift}}_{(D,D),C} \end{pmatrix} = \begin{pmatrix} 1-\alpha & \alpha\gamma \\ \alpha\gamma(1-\alpha) & 1-\alpha+(\alpha\gamma)^2 \end{pmatrix} \begin{pmatrix} Q^t_{(C,C),D} - Q^{\star,\text{lose-shift}}_{(C,C),D} \\ Q^t_{(D,D),C} - Q^{\star,\text{lose-shift}}_{(D,D),C} \end{pmatrix} \ , \tag{107}$$

which yields

$$\begin{pmatrix} u^t \\ v^t \end{pmatrix} = M^t \begin{pmatrix} u^0 \\ v^0 \end{pmatrix} \ , \tag{108}$$

with

$$M = \begin{pmatrix} a & -b \\ -ab & a+b^2 \end{pmatrix} \tag{109}$$

$$a = 1-\alpha \tag{110}$$

$$b = \alpha\gamma \tag{111}$$

$$u^t = Q^t_{(C,C),D} - Q^{\star,\text{lose-shift}}_{(C,C),D} \tag{112}$$

$$v^t = -\left(Q^t_{(D,D),C} - Q^{\star,\text{lose-shift}}_{(D,D),C}\right) \ . \tag{113}$$

The matrix $M$ can be diagonalized:

$$M = \underbrace{\begin{pmatrix} \frac{b^2+\sqrt{4a^2+b^4}}{2ab} & \frac{b^2-\sqrt{4a^2+b^4}}{2ab} \\ 1 & 1 \end{pmatrix}}_{:=P:=(w_1|w_2)} \begin{pmatrix} \lambda_1 & 0 \\ 0 & \lambda_2 \end{pmatrix} \underbrace{\begin{pmatrix} \frac{ab}{\sqrt{4a^2+b^4}} & \frac{1}{2} - \frac{\sqrt{4a^2+b^4}}{2(4a+b)} \\ \frac{-ab}{\sqrt{4a^2+b^4}} & \frac{1}{2} + \frac{\sqrt{4a^2+b^4}}{2(4a+b)} \end{pmatrix}}_{P^{-1}} \ , \tag{114}$$

with $\lambda_1 = a + \frac{b^2+\sqrt{b^4+a^2}}{2}, \lambda_2 = a + \frac{b^2-\sqrt{b^4+a^2}}{2}$.

Let

$$\begin{pmatrix} \delta^0_1 \\ \delta^0_2 \end{pmatrix} := P^{-1} \begin{pmatrix} u^0 \\ v^0 \end{pmatrix} = \begin{pmatrix} \frac{u^0 ab}{\sqrt{4a^2+b^4}} + v^0 \left(\frac{1}{2} - \frac{\sqrt{4a^2+b^4}}{2(4a+b)}\right) \\ \frac{-u^0 ab}{\sqrt{4a^2+b^4}} + v^0 \left(\frac{1}{2} + \frac{\sqrt{4a^2+b^4}}{2(4a+b)}\right) \end{pmatrix} \tag{115}$$

be the decomposition of $\begin{pmatrix} u^0 \\ v^0 \end{pmatrix}$ in the basis $w_1, w_2$:

$$\begin{pmatrix} u^0 \\ v^0 \end{pmatrix} = \delta^0_1 w_1 + \delta^0_2 w_2 \ . \tag{116}$$

Combining Equations (108) and (116) yields

$$\begin{pmatrix} u^t \\ v^t \end{pmatrix} = \delta^0_1 \lambda^t_1 w_1 + \delta^0_2 \lambda^t_2 w_2 \ , \tag{117}$$

and

$$u^t - v^t = \underbrace{\delta_1^0}_{>0} \lambda_1^t \left\langle w_1, \begin{pmatrix} 1 \\ -1 \end{pmatrix} \right\rangle + \underbrace{\delta_2^0}_{>0} \lambda_2^t \left\langle w_2, \begin{pmatrix} 1 \\ -1 \end{pmatrix} \right\rangle \tag{118}$$

$$= \underbrace{\delta_1^0}_{>0} \lambda_1^t \underbrace{\left( \frac{b^2 + \sqrt{4a^2 + b^4}}{2ab} - 1 \right)}_{>0} - \underbrace{\delta_2^0}_{>0} \lambda_2^t \underbrace{\left( 1 - \frac{b^2 - \sqrt{4a^2 + b^4}}{2ab} \right)}_{>0} . \tag{119}$$

Since $\lambda_1 > \lambda_2$, then if $u^{t_1} - v^{t_1} > 0$, then $u^t - v^t > u^{t_1} - v^{t_1} > 0$.

In other words, in order to show that $Q^t_{(C,C),D} - Q^\star_{(C,C),D} + Q^t_{(D,D),C} - Q^\star_{(D,D),C} := u^t - v^t \geq Cste$, then it is sufficient that $u^{t_1} - v^{t_1} \geq Cste$

Hence one only needs to show

$$Q^{t_1}_{(D,D),C} - Q^{t_1}_{(D,D),D} + Q^{t_1}_{(C,C),D} - Q^{t_1}_{(C,C),C} \geq \frac{k\Delta_r \alpha}{1 - \gamma} \tag{120}$$

$$Q^{t_1}_{(C,C),D} - Q^{t_1}_{(C,C),C} \geq \frac{2k\Delta_r \alpha}{1 - \gamma} \tag{121}$$

$$Q^{t_0}_{(C,C),D} - Q^{t_0}_{(C,C),C} \geq \frac{4k\Delta_r \alpha}{1 - \gamma} , \tag{122}$$

$\square$

*Proof.* Lemma D.2 *iii).* Using Equation (116) one has

$$Q^t_{(C,C),D} - Q^{\star,\texttt{lose-shift}}_{(C,C),D} \leq \underbrace{\delta_1^0 \frac{b^2 + \sqrt{4a^2 + b^4}}{2ab}}_{:=C_1} \lambda_1^{t-k} + \underbrace{\delta_2^0 \frac{b^2 - \sqrt{4a^2 + b^4}}{2ab}}_{:=C_2} \lambda_2^{t-k} + \frac{2k\Delta_r \alpha}{1 - \gamma} . \tag{123}$$

$\square$

# E. Additional Experimental Details

Table 3. Prisoner's dilemma rewards parameterization used in the experiments, $1 < g < 2$.

|  | Cooperate | Defect |
|---|---|---|
| Cooperate | 2g  /  2g | 2 + g  /  g |
| Defect | g  /  2 + g | 2  /  2 |

## E.1. Influence of the Incentive to Cooperate $g$

**Comments on Figure 6.**. Figure 6 displays the evolution of $Q$-values difference as a function of the number of iterations in the iterated prisoner's for a self-play evolution of Algorithm 1 in the fully greedy case $\epsilon = 0$. As the incentive to cooperate $g$ increase, the time when the `lose-shift` policy and then the `Pavlov` policy are reached (green zone) decrease, in other words, cooperation is achieved faster.

## E.2. Hyperparameter for the Deep $Q$-Learning Experiments

The hyperparameters used for the deep $Q$-learning experiments are summarized in Table 4. 5 runs are displayed in Figure 5, each run takes 3 hours to train on a single GPU on RTX8000.

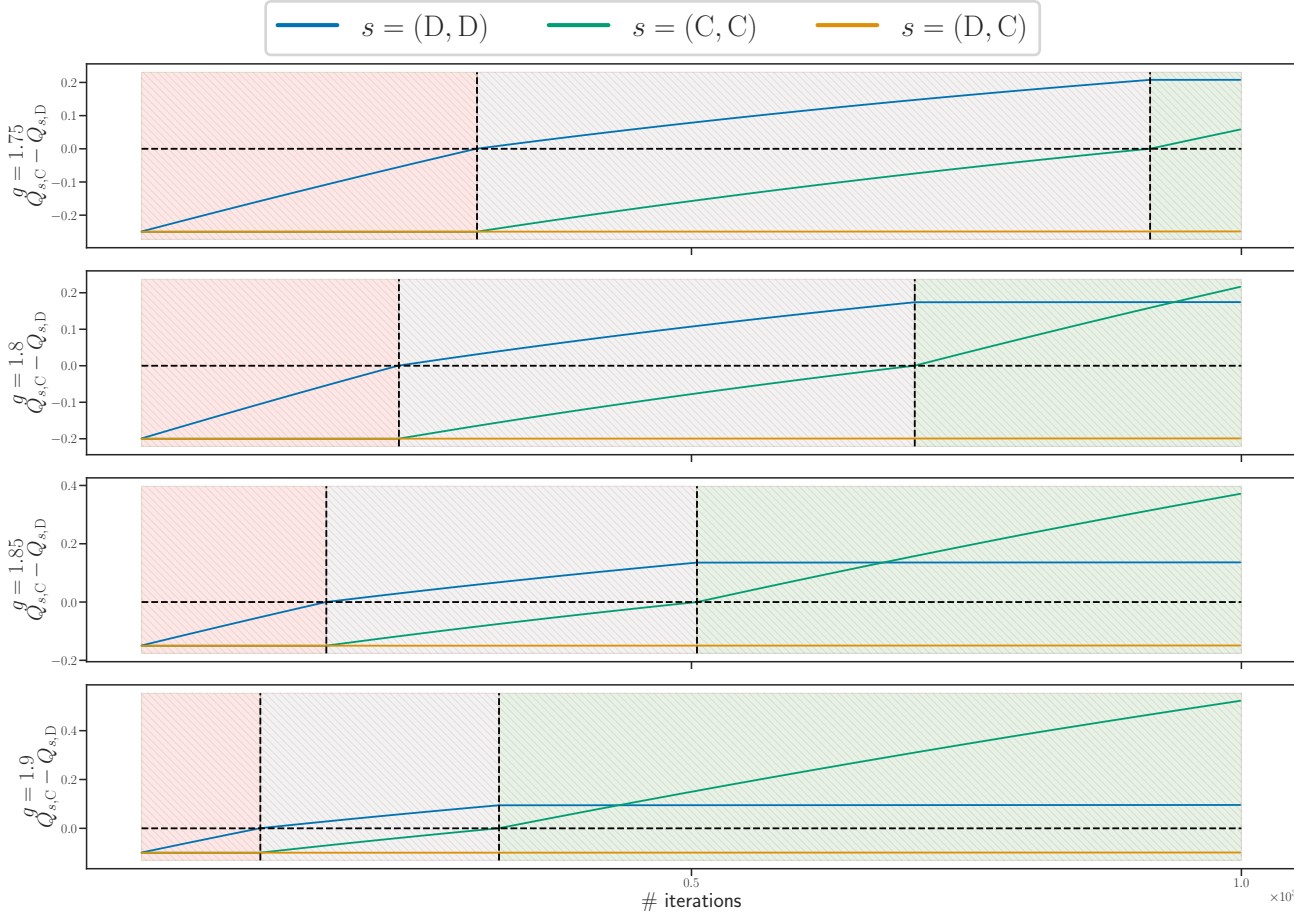

*Figure 6.* **Influence of the incentive to cooperate $g$.** Evolution of the $Q$-learning policy as a function of the number of iterations in the iterated prisoner's dilemma. With a correct (optimistic) initialization, players go from an `always defect` policy to the `lose-shift` policy (at time $t_1$), and then go to the cooperative `Pavlov` policy (at time $t_2$). The discount factor is set to $\gamma = 0.6$. The incentive to cooperate $g$ varies from $g = 1.75$ to $g = 1.9$ (see Table 3).

*Table 4.* List of hyperparameters used in the deep $Q$-learning experiment (Figure 5).

| Hyperparameter | Value |
|---|---|
| tau | 0.01 |
| seed | 8 |
| gamma | 0.8 |
| buffer_capacity | 1000000 |
| decay_eps | true |
| eps_decay_steps | 600 |
| eps_start | 0.5 |
| eps_end | 0.01 |
| loss_type | Huber Loss |
| optimizer_type | SGD |
| hidden_size | 32 |
| num_actions | 2 |
| num_iters | 10000 |
| batch_size | 16384 |
| do_self_play | true |
| pretrain_iters | 600 |
| pretrain_vs_random | true |

