# OpenReview forum: "Self-Play $Q$-Learners Can Provably Collude in the Iterated Prisoner's Dilemma"
_ICML.cc/2025/Conference — ICML 2025 poster_

### Official Review · Reviewer_t2Hj · 2025-03-13

**Overall Recommendation:** 3

**Summary:**

In this work, The authors prove that multi-agent Q-learners playing the iterated prisoner’s dilemma can learn to collude. The complexity of the cooperative multi-agent setting yields multiple fixed-point policies for Q-learning: the main technical contribution of this work is to characterize the  convergence towards a specific cooperative policy. More precisely, in the iterated prisoner’s dilemma, the results show that with optimistic Q-values, any self-play Q-learner can provably learn a cooperative policy called Pavlov, also known as win-stay, lose-shift policy, which strongly differs from the vanilla Pareto dominated always defect policy.

**Claims And Evidence:**

The claims made in the submission are generally well-supported by clear and convincing evidence. The authors provide a thorough theoretical analysis and empirical validation to support their main claims regarding the collusion behavior of multi-agent Q-learners in the iterated prisoner's dilemma.

**Essential References Not Discussed:**

In recent years, there has been significant progress in algorithmic collusion research, but the authors cited less relevant work from the past year (2024) in their literature review.

**Experimental Designs Or Analyses:**

I checked the soundness of the experimental designs and analyses. The experiments are well-designed to test the theoretical claims, specifically focusing on the convergence of Q-learning to the Pavlov policy in the iterated prisoner's dilemma. The authors use a combination of standard Q-learning and deep Q-learning to demonstrate the robustness of their findings. The experimental setup, including the choice of hyperparameters and the initialization process, is clearly described and appears to be valid. The results are presented in a clear and consistent manner, with multiple runs and seeds used to ensure reliability. The parameter selection in the experiment was relatively single, and the impact of different parameter changes on the results was not shown. Overall, the experimental designs and analyses are sound.

**Methods And Evaluation Criteria:**

The proposed methods and evaluation criteria in this submission can solve the problem of collusion in multi-agent Q-learning within the iterated prisoner's dilemma to a certain extent.

**Other Comments Or Suggestions:**

The paper mentions that the self-play assumption is an important premise for proving the convergence of the Q-learning algorithm. However, self-play may lead to agents learning only a single strategy, which may not fully reflect real-world scenarios. In the real world, agents may encounter opponents adopting different strategies. Therefore, if the self-play assumption is neglected, can the algorithm still achieve collusive behavior?

**Other Strengths And Weaknesses:**

Strengths:
1.	The paper provides a novel theoretical analysis of collusion in multi-agent Q-learning, specifically in the iterated prisoner's dilemma. It removes restrictive assumptions from prior work (e.g., memoryless settings) and characterizes the dynamics leading to collusion.
2.	By providing a theoretical foundation for observed empirical phenomena, the paper contributes to the broader discussion on algorithmic collusion and its impact on market competition.

Weaknesses
1.	While the theoretical contributions are strong, the paper could benefit from more discussion on how these findings translate to real-world applications.
2.	The conditions for convergence to the Pavlov policy (e.g., optimistic initialization) may seem restrictive. While the authors provide a detailed analysis, it would be helpful to explore the robustness of these conditions and whether they can be relaxed without compromising the results.
3.	The relevant literature of recent years has not been fully explored; the innovations of the study are not sufficiently prominent.

**Questions For Authors:**

1.	In recent years, there has been significant progress in algorithmic collusion research, but the authors cited less relevant work from the past year in their literature review. It is recommended that the authors add a review of recent literature to provide a comprehensive picture of the latest research advances in the current field to highlight the innovative nature of this study. In addition, the authors should clearly indicate the differences and innovations of this study from existing studies.
2.	The paper demonstrates that the Q-learning algorithm can converge to a collusive strategy (Pavlov strategy). However, the stability of this collusive strategy in the long run has not been discussed, especially in the presence of external disturbances.
3.	The paper presents experimental results showing that the Q-learning algorithm converges to the Pavlov strategy and validates the theoretical analysis. However, the experiments only consider a single payoff parameter 𝑔. To enhance the reliability and generalizability of the results, the authors are encouraged to explore variations in the payoff parameters in the experiments.
4.	The paper focuses on theoretical proofs, but it does not sufficiently address the potential challenges of applying the algorithm in real-world economic scenarios. Additionally, detecting or preventing collusive strategies in practice is an important issue, and the authors are advised to provide more insights on this topic.

**Relation To Broader Scientific Literature:**

The findings relate to economic concerns about algorithmic collusion, extending empirical studies by Calvano et al. (2020) and OECD (2017). The paper provides theoretical evidence that Q-learning agents can learn collusive strategies, contributing to the understanding of tacit collusion in algorithmic pricing. However, the significance of the conclusions obtained in the article in terms of practical applications is not yet clear, and the authors have failed to fully elaborate on their practical value and application prospects. This is a point of improvement in the article.

**Theoretical Claims:**

Yes, I checked the correctness of the proofs for the main theoretical claims, specifically Theorem 3.2 (fully greedy case) and Theorem 3.3 (ϵ-greedy case). The proofs appear to be correct and well-structured. The authors provide detailed derivations and supporting lemmas to demonstrate the convergence of Q-learning to the Pavlov policy. The arguments for the linear convergence rate (Appendix C.2) and the bounds on Q-value deviations (Lemma 3.4) are well-explained and support the main claims effectively.

---

> ### Author Rebuttal · Authors · 2025-04-01
>
> Authors would like to thank Reviewer LKws for their in-depth comments and very insightful review.
>
> > 1 In recent years, there has been significant progress in algorithmic collusion research, but the authors cited less relevant work from the past year in their literature review. It is recommended that the authors add a review of recent literature to provide a comprehensive picture of the latest research advances in the current field to highlight the innovative nature of this study. In addition, the authors should clearly indicate the differences and innovations of this study from existing studies.
>
> We would be happy to incorporate additional references to recent work. Does the reviewer have specific papers in mind that they believe are particularly relevant?
>
> > 2 The paper demonstrates that the Q-learning algorithm can converge to a collusive strategy (Pavlov strategy). However, the stability of this collusive strategy in the long run has not been discussed, especially in the presence of external disturbances.
>
> - In the fully greedy case ($\epsilon = 0$), the $Q$-values ($Q_{CC, D}$ and $Q_{DD, C}$) converge exactly to those corresponding to the Pavlov strategy at equilibrium.
>
> - In the $\epsilon$-greedy case, we demonstrated convergence with high probability. However, there is still a small, exponentially decaying probability (as $\alpha$ increases) that the system may deviate from the Pavlov strategy. Specifically, for a fixed horizon $T$, there exists a set of parameters $\alpha$, $\epsilon$, and $\delta$, such that the Pavlov strategy is reached with probability $1 - \delta$. However, for fixed values of these parameters, there exists a horizon $T$ for which the policy deviates from the Pavlov strategy.
>
> > 3 The paper presents experimental results showing that the Q-learning algorithm converges to the Pavlov strategy and validates the theoretical analysis. However, the experiments only consider a single payoff parameter 𝑔. To enhance the reliability and generalizability of the results, the authors are encouraged to explore variations in the payoff parameters in the experiments.
>
> The trends for multiple values of $g$ were indeed similar, so we initially reported the results for only one value of $g$ for simplicity. However, to enhance clarity and transparency, we have added graphs with additional values of the incentive to cooperate ($g$) in the Appendix of the revised manuscript. Note that $g$ needs to be large enough for the Pavlov and Lose-shift policies to exist (i.e., $g > 4/3$). Additionally, following Reviewer hUYk's suggestion, we plan to remove the $g$ parameterization in favor of a more general reward parameterization.
>
> > 4 The paper focuses on theoretical proofs, but it does not sufficiently address the potential challenges of applying the algorithm in real-world economic scenarios. Additionally, detecting or preventing collusive strategies in practice is an important issue, and the authors are advised to provide more insights on this topic. Practical value and application prospects.
>
> Indeed, we view this work as an initial step toward understanding collusion in more complex environments, such as Bertrand games, and ultimately developing methods for detecting collusion. However, we believe that addressing these practical challenges is particularly challenging and goes beyond the scope of this paper. We hypothesize that increasing the number of players could make collusion more difficult, although providing a full theoretical analysis of this is currently out of reach.
>
> **Action**: We will cite the proposed relevant literature and add additional values of the incentive to cooperate $g$ in the Appendix.

---

### Official Review · Reviewer_J2GP · 2025-03-13

**Overall Recommendation:** 1

**Summary:**

This paper studies whether Q-learning algorithm with self-play and one-step memory can lead to collusion in iterated prisoner's dilemma game. The authors characterize the conditions on the initializations, rewards, and discount factor to guarantee that the agents would shift from always defect to Pavlov strategies.

**Claims And Evidence:**

The paper makes an implicit claim that agents would follow Q-learning algorithm with self-play and one-step memory in the iterated matrix games. However, this claim has not been supported by clear and convincing evidence. Below, I highlight the problematic parts:
-The algorithm structure shows that agents can observe the state and the state is the previous joint actions of the players. This implies that the agents can observe the opponent actions. Then, it is not clear why they construct the Q-function over their local actions only as if they do not observe the opponent actions. Such an approach leads to the claimed Bellman equation (2), which depends on opponent policy \pi^2. (The equation has \pi yet it must be a typo.) This causes ambiguity due to its dependence on opponent policy. The paper resolves this ambiguity with self play. However, the self-play causes further issues:
-- Both agent has the same Q-function estimate all the time. This is made possible through the update of Q-function only for agent 1. For example, agent 2 does not update Q_{s_t,a_t^2} based on its own action a_t^2 and its own reward r_2. Such an update is not justifiable for a non-cooperative environment such as markets.
-- Assumption 3.1 uses different initializations for different state and actions. This is also difficult to justify if the agents do not know the model.
-- If the agents know the model, then it is not justifiable to use Q-learning since they can compute the fixed point directly via dynamic programming.
In summary, the algorithm studied is not well-justified from the multi-agent learning perspective. Therefore, the claims drawn are likely to be induced by the artifacts of this not well-designed algorithm rather than an emerging phenomenon for multi-agent learning algorithms.

**Essential References Not Discussed:**

N/A

**Experimental Designs Or Analyses:**

Use of deep Q-learning does not sound for this simple setup.

**Methods And Evaluation Criteria:**

The paper provides experimental analysis for the iterated prisoner dilemma with specific reward functions. The paper also includes simulations using deep Q-learning. However, it is not clear why neural network approximation would be necessary for such a small scale problem (with four states and 2x2 actions per state) that can be solved in the tabular form.

**Other Comments Or Suggestions:**

In (2), \pi must be \pi^2.

**Other Strengths And Weaknesses:**

The algorithm must be well-justified from a multi-agent learning perspective.
The paper needs to improve its problem formulation and notation.
-For example, the model knowledge and information structure of agents are not clear.
-The notation Q_{s,D}^{*,Defect} has not been introduced. Is it the fixed point of (2) given that \pi=defect? Then, (2) should have been defined accordingly.
-Policies such as always defect, Pavlov, Lose-shift have been used before they get introduced in Table 2 at page 3.
The use of neural networks approximation for such a small scale problem is not clear.

**Questions For Authors:**

- Can the authors explain whether agents can follow such an algorithm in non-cooperative environments? Or is this algorithm for introspective computation of a policy to play in non-cooperative environments?

**Relation To Broader Scientific Literature:**

This is a niche problem for the ICML community. This paper would fit better to more specialized domains such as ACM EC.

**Theoretical Claims:**

I have not checked the correctness of the proofs but they are intuitive given the carefully crafted initialization and the single-sided update of Q-functions. Indeed, I am more concerned on the justification of these assumptions. For example, why would agents use different initializations for different state-action values. Since agents update their actions based on Q-function estimates with "small enough" exploration, carefully crafted initializations can cause bias towards certain action profiles such that the agents play these action profiles only through exploration and therefore they cannot learn their values accurately.

---

> ### Author Rebuttal · Authors · 2025-04-01
>
> > Such an approach leads to the claimed Bellman equation (2), which depends on opponent policy \pi^2. This causes ambiguity due to its dependence on opponent policy.
>
> We are confused by this statement, as dependence on the opponent's strategy is inherent to multi-agent games—the $Q$-values necessarily depend on the opponent's policy.
>
> > The paper resolves this ambiguity with self-play. However, the self-play causes further issues: -- Both agent has the same Q-function estimate all the time. This is made possible through the update of Q-function only for agent 1. For example, agent 2 does not update Q_{s_t,a_t^2} based on its own action a_t^2 and its own reward r_2. Such an update is not justifiable for a non-cooperative environment such as markets.
>
> We strongly push back against this claim. Markets are often modeled as cooperative/competitive environments, such as Bertrand games, where the Prisoner’s Dilemma serves as a minimalistic setting. See Calvano et al., Section 3.
>
> Reference:
>
>  Calvano, Emilio, et al. "Artificial intelligence, algorithmic pricing, and collusion." American Economic
>
> > Assumption 3.1 uses different initializations for different states and actions. This is also difficult to justify if the agents do not know the model. -- If the agents know the model, then it is not justifiable to use Q-learning since they can compute the fixed point directly via dynamic programming.
>
> We would like to push back against this statement: we explicitly provide a practical way to initalize the $Q$-values to satisfy Assumption 3.1. In this case, all the $Q$-values are the same. The reviewer can refer to the **Q-values Initialization in Practice** paragraph following Assumption 3.1 for further clarification.
>
> > The notation $Q_{s,D}^{\\star,\\mathrm{Defect}}$ has not been introduced.
>
>
>
> **$Q_{s,D}^{\star, \mathrm{Defect}} $ is explicitly defined in Proposition 2.1**:
>
> $ Q_{s, D}^{\star, \mathrm{Defect}} =  \mathbb{E}_{a \sim \pi(\cdot | s)}   r\_{D, a} /  (1 -  \gamma)$
>
> $ Q_{s, C}^{\star, \mathrm{Defect}} = Q_{s, D}^{\star, \mathrm{Defect}} -  \mathbb{E}_{a \sim \pi(\cdot | s)}  ( r\_{D, a} -  r\_{C, a})$
>
> $Q^{\star, \mathrm{Defect}} $ is a fixed point of the Bellman Equation that yields an always defect policy in the self-play case.
>
> > Policies such as always defect, Pavlov, Lose-shift have been used before they get introduced in Table 2 at page 3.
>
> Table 2 is now referenced earlier in the text to ensure clarity and avoid confusion
>
> > The use of neural network approximation for such a small-scale problem is not clear.
>
> The idea of this paper is not to provide large-scale deep RL experiments. The goal of the deep Q-learning experiment part is to see how our findings extend to more complex settings. We think it is especially interesting to see how behaviour transfers from a controlled experiment to a larger, more complex setting.
>
> > Typos: In (2), \pi must be \pi^2.
>
> The typos have been fixed
>
> > Can the authors explain whether agents can follow such an algorithm in non-cooperative environments? Or is this algorithm for introspective computation of a policy to play in non-cooperative environments?
>
> We are not sure we understand this question. What exactly are you referring to with "introspective" and "our algorithm"?

---

### Official Review · Reviewer_hUYk · 2025-03-13

**Overall Recommendation:** 3

**Summary:**

The paper studies Q-learning in the Iterated Prisoner’s Dilemma with memory 1. It shows by formal proof that under some condition, Q-learning results in the so-called Pavlov strategy, which forms a cooperative equilibrium. The paper also conducts some experiments, including experiments with Deep Q-learning on the same Iterated PD.


## update after rebuttal

Given that this paper received mixed scores and that I know this area reasonably well and am quite interested in it, I spent relatively much time looking at the paper again during rebuttal.

I'm sticking with my "weak accept" recommendation. If anything, I'm now torn between "weak accept" and "accept". I think this is a solid paper and it should be accepted.

That said, my recommendation is made under the assumption that the authors will use the extra page in the camera-ready version to clarify some things and better relate the paper to prior findings with somewhat different results. See my previous review, my response to the rebuttal, and the below comments for details.


More in-the-weeds comments from re-reading some parts:

I looked at the proof of Theorem 3.2 again in some detail to better understand the underlying dynamics.

It seems to me that self-play is at least somewhat important for this step of the proof. In particular, when you switch to playing C in (D,D), it seems important that your opponent also switches at the same time. This way, you will now learn that C in (D,D) is quite good (it gets you the (C,C) payoff), even though from a "counterfactual" perspective it's actually worse (if you could switch to D while having your opponent stick to C, that would be better). If the time of switching from D to C was very out of sync (because the two players have separate Q functions, etc.), then once the first player switches to C, they would very quickly learn that C is even worse than D, while the other player would start receiving more positive results again with D, because they would get the (D,C) "temptation" payoff.

In principle, this dynamic used in the proof even applies to training in a one-shot Prisoner's Dilemma: If you switch very discontinuously from (almost) always always defecting to (almost) always cooperating ((epsilon-)greedy) and you train via self-play, then cooperation is (somewhat) stable, because you only (mostly) ever cooperate when your opponent also cooperates. IIRC, these kinds of dynamics were studied by https://proceedings.neurips.cc/paper/2021/file/b9ed18a301c9f3d183938c451fa183df-Paper.pdf I think this says that in the one-shot PD setup with self-play, epsilon-greedy allows convergence to mostly cooperating (in terms of frequencies, not iterates) due to this dynamic or at least their results don't rule it out. Generally the self-play setting of the present paper can be viewed as a special case of the setting of that paper (I think). But I think the only result from that paper that somewhat directly applies to this paper just says that one can only converge to Nash equilibria.



And another request for clarification: In the Deep Q-learning setup in Section 5, is there anything akin to optimistic initialization happening? (More minor: "As discussed in the previous paragraph" -- I don't understand which part of the previous paragraph this refers to. Maybe a paragraph was deleted here at some point?)

**Claims And Evidence:**

Yes.

**Essential References Not Discussed:**

There are more papers on learning agents in the iterated Prisoner’s Dilemma or very similar games. (The authors already cite some of these papers.)

I’d be especially interested in a discussion of related papers that have found something more like the opposite result, i.e., that training typically doesn’t find cooperative strategies. E.g., https://www.sciencedirect.com/science/article/pii/0303264795015515
https://arxiv.org/pdf/1709.04326 https://arxiv.org/abs/2211.14468 https://arxiv.org/pdf/1811.08469 https://arxiv.org/abs/2012.02671 All of these papers use somewhat different methods on somewhat different games. But they all find that if one just does some kind of “naive best response” learning, one doesn’t learn to cooperate. I’d like to know what the relevant differences are to this paper. E.g., is it the slightly increased complexity of the environment? Or not using self-play?

Somewhat relatedly (and less importantly): Quite a few papers have proposed methods for finding more cooperative equilibria. In principle, the results in the paper could also be used to this end. This is very briefly discussed in Sect. 3.1. It would be interesting to see a more detailed discussion of how this would relate to these other methods, though I think the authors view their project mostly as descriptive rather than prescriptive. So, perhaps it’s not that interesting to talk about how various other methods for achieving cooperation are better/worse.

**Experimental Designs Or Analyses:**

I didn’t check anything beyond what’s written in the main text.

**Methods And Evaluation Criteria:**

Yes.

**Other Comments Or Suggestions:**

In cases like the below, it’s nicer, in my view, to give seminal rather than textbook references.

>When the prisoner’s dilemma is infinitely repeated, new equilibria can emerge, and always defect is no longer the dominant strategy (Osborne, 2004).

>A popular approach to maximize the cumulative reward function Equation (1) is to find an action-value function or Q-function, that is a fixed point of the Bellman operator (Sutton and Barto, 2018, Eq. 3.17).


>the usual way to deal with multiple agents in reinforcement learning applications (Lowe et al., 2017; Silver et al., 2018; Baker et al., 2019; Tang, 2019).

It might be worth noting here that this requires symmetry of the game. Also, is this really the usual way to deal with multiple agents in _general-sum_ games?

Why not write the second math line in the Prop. 2.1 in the same way as the first? Isn’t that shorter and easier to read?

>The main takeaway from Proposition 2.2 is that there exists a fixed point of the Bellman Equation (2) whose associated strategy is cooperative. Interestingly, tit-for-tat

It would be nice to say something here about the relation between SPE and being a fixed point.

>In this section, we show convergence of the dynamics resulting from ϵ-greedy Q-learning with memory updates (i.e., Equation (3)) toward the cooperative fixed point Pavlov policy.

The next sentence spells it out, but this sentence alone reads quite weird. Obviously, you can also converge to the “always defect” policies.

>distribution ρ over the initial state space S

This is a bit awkward, because S is just the regular state space, right?

In Theorem 3.2, I find this a little confusing:
>Suppose the initial policy is always defect
The version of Q-learning considered here doesn’t take a policy as input. Is this implicitly a constraint on the Q values?

Relatedly in the proof: Why is $Q_{s,D} > Q_{s,C}$ in Phase 1?  This doesn’t follow from Assumption 3.1, right?

In Algorithm 1: I assume s_t is updated and Alg. 2 is called with the current s_t?

>As opposed to the vanilla Q-learning cases, in which the agents learn to cooperate is not the same, but the resulting policy is.

I’m unable to parse this sentence.

In the references:
>Competition Bureau. Big data and innovation: Implications for competition policy in canada.
Canada should be capitalized!

**Other Strengths And Weaknesses:**

I like how the paper uses theoretical perspectives (talk of subgame-perfect equilibrium), while also engaging with the practice of multi-agent ML.

I haven’t checked the proofs, but I’ve thought about this sort of issue enough to understand that it is difficult and very tedious to prove this sort of result. So, although the contribution could be viewed as small (in that it is about a specific, simple game, etc.), the amount of effort necessary to obtain this sort of result is, I believe, quite large.

The question of whether/when cooperation can emerge in RL is important.

In my mind, the main weakness of the paper is that it is a little hard to understand, both at the micro level – see the detailed suggestions below – and at the level of motivation and interpretation for the results. From more to less central:

1) Since Pavlov is a fixed point of the Q function, it’s, I assume, relatively easy to show that there are some initializations under which the Q-learner starts out and sticks with Pavlov. (Right? E.g., take whatever Q values you get in Phase 3 and just initialize it to those.)

I take it that the paper is interesting in large part because it doesn’t just do this. Instead, it considers initializations that are quite far from Pavlov and show that there’s a robust-enough-to-prove path from these initializations to Pavlov. It gives more interesting starting conditions under which the players learn to play Pavlov.

But then the paper doesn’t really say all that much about these conditions, about why they’re more interesting than the trivial conditions, etc. Even the characterization of the Q values as “optimistic” isn’t really explained.
2) How was Table 1 picked. It’s natural to pick a single-parameter version of the PD (rather than a fully parameterized one). For instance, a simple one is: playing D gives you 1 and cooperating gives the other player y>1. but the specific parameterization is quite odd. What does g control, intuitively? Also, wouldn’t it be good to be able to arbitrarily control the ratio of gains from cooperation to gains from defection? (y in the above) That way one could think about what happens if this ratio goes to infinity?

**Questions For Authors:**

1. See item 1 from the “Other Strengths And Weaknesses” section.
2. See item 2 from the “Other Strengths And Weaknesses” section.
3. (The second paragraph from “Essential References Not Discussed” also asks questions, but not ones that I expect the authors to answer in the short amount of available time.)

**Relation To Broader Scientific Literature:**

There are a bunch of papers on whether agents trained with RL or the like can learn to cooperate in various Prisoner’s Dilemma-type games. I mostly view this paper as a contribution to this literature.

**Theoretical Claims:**

I tried to read the proof of Theorem 3.2, but didn’t verify the details.

---

> ### Author Rebuttal · Authors · 2025-04-01
>
> Authors would like to thank Reviewer hUYk for their in-depth comments, which have significantly improved the updated version of the manuscript.
>
> > I’d like to know what the relevant differences are to this paper. E.g., is it the slightly increased complexity of the environment? Or not using self-play?
>
> The key distinction between our work and all the papers cited in the review is that they adopt an empirical approach, mostly introducing new algorithms/settings to promote cooperation. For ourselves, we establish theoretical results on achieving cooperation for an existing algorithm: $Q$-learning. Below, we clarify the differences between our algorithm and those considered in the cited papers, as well as why we believe they report divergent findings (i.e., that cooperation is difficult to achieve).
>
> - **Sandholm and Crites, 1996**: This study examines the same setting as ours but employs a policy gradient method instead of Q-learning and does not consider self-play. Given their experimental findings, our theoretical results are somewhat surprising. Notably, empirical evidence suggests that relaxing the self-play assumption still leads to cooperation “most of the time” (see Barfuss and Meylahn, 2023, Fig. 1a). **Our perspective**:
>   - In retrospect, we believe their findings differ because policy gradient methods struggle to discover cooperative policies.
>   - Barfuss and Meylahn suggest that the absence of self-play does not inherently prevent cooperation from emerging.
>
> - **Foerster et al., 2018; Letcher et al., 2019**: These works build on Sandholm and Crites (1996) by introducing opponent shaping, where updates account for the opponent’s learning process using first-order methods. Their approach aims to address the shortcomings of standard policy gradient methods. While they apply their methods to slightly more complex games, tabular Q-learning would still be feasible in such settings.
> **Our perspective**:
>    - Their conclusions differ from ours because they focus on policy gradient methods, which, as shown by Sandholm and Crites (1996), struggle to find cooperative policies.
>    - They consider a slightly larger game (the coin game) and relax the self-play assumption. It would be interesting to test whether tabular Q-learning can still identify cooperative strategies in this setting.
>
> - **Oesterheld et al., 2023**:
>  This work investigates the one-shot Prisoner’s Dilemma, a setting where cooperation is known to be unattainable in equilibrium. They circumvent this limitation by introducing transparency, allowing agents to share information about their similarities. While their problem setup is more constrained than ours, it introduces distinct challenges.
>
> - **Hutter et al., 2020**:
>  This study integrates ideas from Foerster et al. (2018) and Letcher et al. (2019), along with transparency mechanisms similar to those in Oesterheld et al. (2023). Thus, their conclusions diverge from ours for similar reasons.
>
> > 1 Since Pavlov is a fixed point of the Q function, it’s, I assume, relatively easy to show that there are some initializations under which the Q-learner starts out and sticks with Pavlov.
>
> Yes Indeed! If the Q-values are initialized exactly at one of the equilibria, e.g., the Q-values corresponding to the Pavlov policy, then the policy sticks to Pavlov. This is also true for the always defect policy.
>
>  *The paper's goal is to show that one can initialize the Q-values such that the initial policy is always defect*, but still converge toward a cooperative policy. This is not obvious because one has to show convergence towards a specific equilibrium, which is non trivial.
>
>
>
> > 2) How was Table 1 picked?
>
> This table comes from Bancio and Mantegazza, a work we directly build upon; that´s why we chose this parameterization to begin. Intuitively, $g$ controls the incentive to cooperate; the larger the $g$, the larger the cooperation is incentivized. In this setting, $g$ can vary between 1 and 2. We would like to stress that all the results are derived with general rewards $r_{a^1, a^2}$ (as it can be seen in Appendicies B, C or D) and the 1D parameterization is mostly used for clarity of the presentation (e.g., the existence of the Pavlov policy, Prop. 2.2). We will remove this parameterizationand keep the general reward formulation $r_{a^1, a^2}$.
>
> **Action**: We clarified these discussions in the manuscript and adopted the general reward parameterization. All the comments from the Comments or Suggestions section have been addressed in the updated manuscript, however, we chose not to answer them for the conciseness of the rebuttal.

---

> > ### Comment · Reviewer_hUYk · 2025-04-04
> >
> > I thank the authors for the rebuttal. While I still have some concerns and plan to look at the paper again and weigh all the considerations, I am for now increasing my score to Weak Accept.
> >
> >
> > Responses:
> >
> > Thanks for relating your paper to these prior papers. In my mind, the contrast to the general sense from these other papers that reciprocal cooperation doesn't just emerge spontaneously without some kind of nudging (like opponent shaping) is very interesting about this paper, so I'd recommend highlighting it more. Also, for those who use citation graphs, it would definitely be helpful for this paper to cite as many papers as possible that somewhat centrally make these kinds of claims. (There are lots of papers making this point. E.g., here's another one: https://arxiv.org/pdf/1806.04067 ) Certainly, I would have liked to read this paper right after reading, say, the Foerster et al. 2018 one.
> >
> > Regarding the nitty-gritty of why the results are different: Obviously, it's a little out of scope for this paper to discuss the setups of other papers. Maybe it's something for future work instead. But I'd be very interested in reading more about this, specifically w.r.t. the most similar papers. (Not sure which one of the ones I mentioned is most similar. Probably one of the papers that specifically consider the IPD. I suppose the Hutter paper has an even simpler policy space, but one that doesn't allow for anything analogous to Pavlov.) Especially if the authors basically already know and would just need to explain. I don't find the theory about gradients immediately intuitive.
> >
> > Re 1: Thanks for this. This is what I figured, but it would be good to be clearer in the paper about why the condition in the paper is particularly interesting.
> >
> > Re 2: I welcome using a more generic / simpler parameterization.

---

### Official Review · Reviewer_LKws · 2025-03-14

**Overall Recommendation:** 4

**Summary:**

This paper shows that in the Prisoner's dilemma, Q-learning agents can learn to collude to a collaborative policy.
The authors clearly identify the underlying assumption to such behaviours.

**Claims And Evidence:**

The main contributions of the paper is to prove that both with (Theorem 3.3) and without exploration (Theorem 3.2), agents can learn a cooperative policy.
Proofs and proof outlines are provided for the main results, which seem to be correct.

**Essential References Not Discussed:**

Related works are discussed at a suitable length.

**Experimental Designs Or Analyses:**

The experimental analysis appears to be correct. This is most a theory paper, experiments are used to back the theoretical claims in the case of deep Q-learning.

**Methods And Evaluation Criteria:**

Yes, the methods are sound and the experimental evaluation appropriate.

**Other Comments Or Suggestions:**

The Pavlov policy has long been known as an optimal policy for the repeated prisoner's dilemma. What is nice in this paper is the proof that this policy can actually be learnt by using Q-learning.
This is interesting and useful information, as there exist several strategies which are fixed points of the Bellman equation and could theoretically be learnt by Q-learning, as the authors point out. This paper shows convergence to a specific equilibrium.

**Other Strengths And Weaknesses:**

This paper shows an interesting convergence result of Q-learning in the prisoner's dilemma, which might find applications in markets.
Still, the contributions is mainly theoretical. Also, a number of assumptions are made on initial conditions and the behaviour of agents. So, the results are rather tailored to the particular setting in hand.
It would be interesting if the authors discussed the broader implications of their results.

**Questions For Authors:**

1) The authors prove the converge results for the case of one-step memory.
How is this case motivated?

2) In the related work section, they also discuss related results for the memoryless case.
But what would change if we assume perfect memory?

3) The authors provide their results based on Assumption 3.1 and motive such assumption. Nonetheless, do the authors have any intuition about the behaviour of Q-learning for other initial conditions.

**Relation To Broader Scientific Literature:**

As opposed to (Banchio and Skrzypacz, 2022; Banchio and Mantegazza, 2022), the authors consider the standard stochastic (i.e., not averaged) version of \epsilon-greedy Q-learning with memory.

The paper cites (Banchio and Mantegazza, 2022) at various times, as there converge of Q-learning in the prisoner's dilemma is considered for the memoryless case.
Still, the results obtained are quite distinct from (Banchio and Mantegazza, 2022) -- where a cooperative strategy does not exists -- to justify a separate analysis and contribution.

**Theoretical Claims:**

Proofs of the main results are provided in the body of the paper. They appear to be correct.
Proofs of other results are given in the appendix. I did not check them.

---

> ### Author Rebuttal · Authors · 2025-04-01
>
> Authors would like to thank Reviewer LKws for their in-depth comments and very insightful review.
>
> > 1 - The authors prove the convergence results for the case of one-step memory. How is this case motivated?
>
>
> The primary motivation for this work stems from the findings of Banchio and Mantegazza (2022), who showed that in the memoryless case with averaged updates, cooperative equilibria do not emerge—agents consistently learn the always defect equilibrium, or an equilibrium residing at the boundary of the cooperate-defect regions (see Figure 5 of their paper). We hypothesized that this lack of cooperation was specific to the no-memory setting and that introducing even a minimal, one-step memory could facilitate convergence toward cooperation. Our results confirm this hypothesis.
>
> Reference:
>  Banchio, Martino, and Giacomo Mantegazza. 2022. "Artificial Intelligence and Spontaneous Collusion."
>
> > 2 - In the related work section, they also discuss related results for the memoryless case. But what would change if we assume perfect memory?
>
> The choice to focus on one-step memory is motivated both conceptually and practically. Axelrod (1980a, 1980b) demonstrated in his repeated Prisoner's Dilemma tournament that the most effective human-crafted strategy (TIT FOR TAT) relied on short-term memory, was forgiving, and was the simplest among the submitted policies. This suggests that minimal memory is sufficient for cooperative behavior to emerge.
>
>
>
> From a theoretical point of view, going beyond the one-step memory is significantly much more complex to analyze and quickly becomes computationally intractable: for instance, with a two-step memory, the state space is of size $2^4 = 16$, which yields a number of $2^{2^4} > 6 \cdot 10^4$ possible policies, which would be significantly more challenging to analyze theoretically (and potentially for only marginal conceptual gains).
>
> From a practical point of view, reinforcement learning models for the Iterated Prisoner's Dilemma often employ GRUs or LSTMs (e.g., in LOLA [Foerster et al., 2018]), but these architectures are still constrained in their memory capabilities—LSTMs, for instance, can typically recall only up to 150 steps (Khandelwal et al., 2018). Similarly, transformer-based approaches are limited by context window length. However, such deep-learning considerations fall outside the scope of our study.
>
>
> Axelrod, Robert. "More effective choice in the prisoner's dilemma." Journal of conflict resolution, 1980a
> Axelrod, Robert. "Effective choice in the prisoner's dilemma." Journal of conflict resolution, 1980b
>
> Foerster, Jakob N., et al. "Learning with opponent-learning awareness." AAMAS 2018
>
> Khandelwal, Urvashi, et al. "Sharp nearby, fuzzy far away: How neural language models use context." arXiv preprint arXiv:1805.04623 (2018).
>
>
> > 3 The authors provide their results based on Assumption 3.1 and motive such assumption. Nonetheless, do the authors have any intuition about the behaviour of Q-learning for other initial conditions.
>
> Yes! We have clear intuitions about the behavior of Q-learning under different initial conditions. For instance,
> - If Assumption i) is not satisfied, then one can show that the agents learn the always defect policy. As for the proof of Thm 3.2, one can study the evolution of Equation (4), and show that $Q_{(DD), D}^t$ converges to $Q_{(DD), D}^{\text{always defect}}$, and always stay larger than $Q_{(DD), C}^t = Q_{(DD), C}^{t_0}$, hence the policy remains always defect. This is also illustrated on the left plot of Figure 3.
> - If Assumption i) holds but Assumption ii) does not, the learned policy follows a lose-shift strategy. This can be established using similar arguments by analyzing the evolution of the system of Equations (5-6).
>
> **Action:**  These discussions will be added to the revised manuscript.

---

### Decision · Program_Chairs · 2025-05-01

**Decision:**

Accept (poster)

**Comment:**

This submission received four expert reviews, with an initial disagreement on whether the paper properly addresses the multi-agent setting that it studies. All reviewers acknowledged the rebuttal and most engaged in a very lively and creative discussion about it. After receiving the authors' responses on multiple queries, the panel concluded that the paper will make a useful contribution to the MARL literature provided that some ambiguities are clarified which is expected to be the case for the camera-ready. The main confusion stemmed from the way in which the initial version of the paper motivated and, subsequently, resolved the tension between independent MARL, collusion incentives and self-play. In the process, some other minor issues were fixed (e.g., missing assumptions in certain statements) which were also addressed during the reviewing period.

In sum, the paper benefited from useful comments and suggestions from four expert reviewers who went above and beyond to discuss the paper and reach a fair evaluation. While the initial version had some important flaws, mainly in presentation, the panel is confident that the post-review version addresses these issues and will make a solid contribution - which was already present in the initial, submitted version - to the MARL literature.